# Near Optimal Reconstruction
# of Spherical Harmonic Expansions

**Amir Zandieh**
Independent Researcher
amir@zed512.gmail.com

**Insu Han**
Yale University
insu.han@yale.edu

**Haim Avron**
Tel Aviv University
haimav@tauex.tau.ac.il

## Abstract

We propose an algorithm for robust recovery of the spherical harmonic expansion of functions defined on the $d$-dimensional unit sphere $\mathbb{S}^{d-1}$ using a near-optimal number of function evaluations. We show that for any $f \in L^2(\mathbb{S}^{d-1})$, the number of evaluations of $f$ needed to recover its degree-$q$ spherical harmonic expansion equals the dimension of the space of spherical harmonics of degree at most $q$, up to a logarithmic factor. Moreover, we develop a simple yet efficient kernel regression-based algorithm to recover degree-$q$ expansion of $f$ by only evaluating the function on uniformly sampled points on $\mathbb{S}^{d-1}$. Our algorithm is built upon the connections between spherical harmonics and Gegenbauer polynomials. Unlike the prior results on fast spherical harmonic transform, our proposed algorithm works efficiently using a nearly optimal number of samples in any dimension $d$. Furthermore, we illustrate the empirical performance of our algorithm on numerical examples.

## 1 Introduction

We consider the fundamental problem of recovering a function from a finite number of (noisy) observations. To provide accurate and reliable predictions at unobserved points we need to avoid overfitting which is typically achieved through restricting our estimator or interpolant to a family of *smooth or structured* functions. In this paper, we focus on interpolating square-integrable functions on the $d$-dimensional unit sphere, with low-degree spherical harmonics, a critical task in scenarios where rotational invariance is a fundamental property. Spherical harmonics are essential in various theoretical and practical applications, including the representation of electromagnetic fields [Wei95], gravitational potential [Wer97], cosmic microwave background radiation [KKS97] and medical imaging [CLL15], as well as modelling of 3D shapes in computer graphics [KFR03] and computer vision [GCL$^+$22]. Further notable real-life applications include molecular/atom systems, where understanding the underlying functions within a spherical context can significantly enhance predictive modeling and simulation accuracy [EEHM17, LWL$^+$21, ZDK$^+$22].

We begin by observing that any function $f$ in $L^2(\mathbb{S}^{d-1})$, i.e., the family of square-integrable functions on the sphere $\mathbb{S}^{d-1}$, can be uniquely decomposed into orthogonal spherical harmonic components. Specifically, if we denote the space of spherical harmonics of degree $\ell$ in dimension $d$ by $\mathcal{H}_\ell(\mathbb{S}^{d-1})$, any $f \in L^2(\mathbb{S}^{d-1})$ has a unique orthogonal expansion $f = \sum_{\ell=0}^\infty f_\ell$ with $f_\ell \in \mathcal{H}_\ell(\mathbb{S}^{d-1})$ (Lemma 2). With this observation, we aim to find the best spherical harmonic approximation of degree $\leq q$ to $f$ using minimal number of samples (essentially treating higher order terms in $f$'s expansion as noise).

37th Conference on Neural Information Processing Systems (NeurIPS 2023).

**Problem 1** (Informal Version of Problem 2). *For an unknown function $f \in L^2(\mathbb{S}^{d-1})$ and an integer $q \geq 1$, efficiently (both in terms of number of samples from $f$ and computations) learn the first $q + 1$ spherical harmonic components $\{f_\ell \in \mathcal{H}_\ell(\mathbb{S}^{d-1})\}_{\ell=0}^{q}$ of $f$ which minimize*

$$\left\| \sum_{\ell=0}^{q} f_\ell - f \right\|_{\mathbb{S}^{d-1}}^{2} := \int_{\mathbb{S}^{d-1}} \left| \sum_{\ell=0}^{q} f_\ell(w) - f(w) \right|^2 dw. \tag{1}$$

The *angular power spectrum* of $f$ commonly obeys a power law decay of the form $\|f_\ell\|_{\mathbb{S}^{d-1}}^2 \leq \mathcal{O}(\ell^{-s})$, for some $s > 0$, depending on the order of differentiability of $f$. In fact, for any infinitely differentiable $f$, $\|f_\ell\|_{\mathbb{S}^{d-1}}^2$ decays asymptotically faster than any rational function of $\ell$. Furthermore, for any real analytic $f$ on the sphere, $\|f_\ell\|_{\mathbb{S}^{d-1}}^2$ decays exponentially. Thus, the first $q + 1$ spherical harmonic components of $f$ typically well approximate $f$ for even modest $q$, and answering Problem 1 is meaningful for a wide range of differentiable functions.

## 1.1 Our Main Results

We reformulate Problem 1 as a least-squares regression and then solve it using randomized numerical linear algebra techniques. We first consider an orthonormal projection operator that maps functions in $L^2(\mathbb{S}^{d-1})$ onto the space of bounded-degree spherical harmonics $\bigoplus_{\ell=0}^{q} \mathcal{H}_\ell(\mathbb{S}^{d-1})$. Specifically, if $\mathcal{K}_d^{(q)}$ is an operator that maps any $f$ with expansion $f = \sum_{\ell=0}^{\infty} f_\ell$ where $f_\ell \in \mathcal{H}_\ell(\mathbb{S}^{d-1})$, onto $f$'s first $q + 1$ expansion components, i.e., $\mathcal{K}_d^{(q)} f = \sum_{\ell=0}^{q} f_\ell$, then Problem 1 can be formulated as

$$\min_{g \in L^2(\mathbb{S}^{d-1})} \left\| \mathcal{K}_d^{(q)} g - f \right\|_{\mathbb{S}^{d-1}}^2.$$

However, solving this regression problem with "continuous" cost function is challenging. To avoid such continuous optimizations, we adopt the approach of [AKM+19] which discretizes the aforementioned regression problem according to the leverage scores of the operator $\mathcal{K}_d^{(q)}$. It turns out that if we can draw random samples with probabilities proportional to the leverage scores of $\mathcal{K}_d^{(q)}$ then we can recover the degree-$q$ spherical harmonic expansion of $f$, i.e. $\sum_{\ell=0}^{q} f_\ell$, with finite number of observations. Particularly, by exploiting the connections between spherical harmonics and *Zonal (Gegenbauer) Harmonics* and the fact that zonal harmonics are the reproducing kernels of $\mathcal{H}_\ell(\mathbb{S}^{d-1})$ (Lemma 3), we prove that the leverage scores of $\mathcal{K}_d^{(q)}$ are constant everywhere on the sphere $\mathbb{S}^{d-1}$. Thus, solving a discrete regression problem with uniformly sampled observations yields a near-optimal solution to Problem 1. Informal statements of our results are as follows.

**Theorem 1** (Informal Version of Theorem 5). *Let $\beta_{q,d}$ be the dimension of spherical harmonics of degree at most $q$, i.e., $\beta_{q,d} \equiv \dim\left(\bigoplus_{\ell=0}^{q} \mathcal{H}_\ell(\mathbb{S}^{d-1})\right)$. There exists an algorithm that finds a $(1 + \varepsilon)$-approximation to the optimal solution of Problem 1, given $s = \mathcal{O}(\beta_{q,d} \log \beta_{q,d} + \varepsilon^{-1} \beta_{q,d})$ observations of function $f$ at uniformly sampled points on $\mathbb{S}^{d-1}$, with $\mathcal{O}(s^2 d + s^\omega)$[1] runtime.*

We also prove that our bound on the number of required samples is optimal up to a logarithmic factor.

**Theorem 2** (Informal Version of Theorem 6). *Any (randomized) algorithm that takes $s < \beta_{q,d}$ samples on any input fails with probability greater than $9/10$, where $\beta_{q,d} \equiv \dim\left(\bigoplus_{\ell=0}^{q} \mathcal{H}_\ell(\mathbb{S}^{d-1})\right)$.*

## 1.2 Related Work

Efficient reconstruction of functions as per Problem 1 has been extensively studied in various fields. Many prior papers considered reconstructing 1-dimensional functions from finite number of samples on a finite interval under smoothness assumption about the underlying function. Notably, the influential line of work of [SP61, LP61, LP62, XRY01] focused on reconstructing Fourier-bandlimited functions and [CKPS16, KVZ19, EMM20, BKM+23] considered interpolating Fourier-sparse signals. Recently, [AKM+19] unified these reconstruction methods in dimension $d = 1$ and gave a universal sampling framework for reconstructing nearly all classes of functions with Fourier-based smoothness constraints. One can view 1-dimensional functions on a finite interval as functions on the unit circle

---

[1]$\omega < 2.3727$ is the exponent of the fast matrix multiplication algorithm [Wil12]

$\mathbb{S}^1$. Thus, Problem 1 is indeed a generalization of the aforementioned prior work to high dimensions under the assumption that the *generalized Fourier series* (Lemma 2) of the underlying function only contains bounded-degree spherical harmonics. This degree constraint on spherical harmonic expansions can be viewed as the $d$-dimensional analog of Fourier-bandlimitedness on circle $\mathbb{S}^1$.

Computing the spherical harmonic expansion in dimension $d = 3$ has received considerable attention in physics and applied mathematics communities. The algorithms for this special case of Problem 1 are known in the literature as "fast spherical harmonic transform" [SS00, ST02]. Most notably, [RT06] proposed an algorithm for computing spherical harmonic expansion of degree $\leq q$ to precision $\varepsilon$ using $\mathcal{O}(\beta_{q,3})$ samples and $\mathcal{O}(\beta_{q,3} \log \beta_{q,3} \cdot \log(1/\varepsilon))$ time. These fast algorithms were developed based on fast Fourier and associated Legendre transforms and make use of a (well-conditioned) orthogonal basis for $\mathcal{H}_\ell(\mathbb{S}^{d-1})$, which happened to be the associated Legendre polynomials when $d = 3$. However, it is in general very difficult to compute an orthogonal basis for spherical harmonics [MNY06], so unlike our Theorem 1, it is inefficient to extend these prior results to higher $d$.

From the techniques point of view, a related work is [GMMM21], which employs harmonic analysis over $\mathbb{S}^{d-1}$ to analyze the generalization of two-layered neural tangent kernels. They show that an unknown function defined on $\mathbb{S}^{d-1}$ can be efficiently recovered using kernel regression w.r.t. neural tangent kernel on uniform random samples from the function. However, the number of samples that [GMMM21] requires for recovering bounded degree spherical harmonics, especially when the degrees are high, is sub-optimal and is strictly worse than our result. Additionally, [GMMM21] does not guarantee recovery with relative error, while our Theorem 5 provides relative error guarantees.

Furthermore, recent applications of Gegenbauer polynomials, along with other orthonormal polynomials like Hermite polynomials, have been found in designing efficient random features for approximating various kernel functions. These applications extend to dot-product kernels [HZA22], Neural Tangent Kernels [ZHA+21, HZL+22], and Gaussian kernels [WZ22, AKK+20].

## 2    Mathematical Preliminaries

We denote by $\mathbb{S}^{d-1}$ the unit sphere in $d$ dimension. We use $|\mathbb{S}^{d-1}| = \frac{2\pi^{d/2}}{\Gamma(d/2)}$ to denote the surface area of sphere $\mathbb{S}^{d-1}$ and $\mathcal{U}(\mathbb{S}^{d-1})$ to denote the uniform probability distribution on $\mathbb{S}^{d-1}$. We denote by $L^2\left(\mathbb{S}^{d-1}\right)$ the set of all square-integrable real-valued functions on sphere $\mathbb{S}^{d-1}$. Furthermore, for any $f, g \in L^2\left(\mathbb{S}^{d-1}\right)$ we use the following definition of the inner product on the unit sphere[2],

$$\langle f, g \rangle_{\mathbb{S}^{d-1}} := \int_{\mathbb{S}^{d-1}} f(w)g(w)dw = \left|\mathbb{S}^{d-1}\right| \cdot \mathbb{E}_{w \sim \mathcal{U}(\mathbb{S}^{d-1})}[f(w)g(w)]. \tag{2}$$

The function space $L^2\left(\mathbb{S}^{d-1}\right)$ is complete with respect to the norm induced by the above inner product, i.e. $\|f\|_{\mathbb{S}^{d-1}} := \sqrt{\langle f, f \rangle_{\mathbb{S}^{d-1}}}$, so $L^2\left(\mathbb{S}^{d-1}\right)$ is a *Hilbert space*.

We often use the term *quasi-matrix* which is informally defined as a "matrix" in which one dimension is finite while the other is infinite. A quasi-matrix can be *tall* (or *wide*) meaning that there is a finite number of columns (or rows) where each one is a functional operator. For a more formal definition, see [SA22].

*Spherical Harmonics* are the solutions of Laplace's equation in spherical domains and can be thought of as functions defined on $\mathbb{S}^{d-1}$ employed in solving partial differential equations. Formally,

**Definition 1** (Spherical Harmonics)**.** *For integers $\ell \geq 0$ and $d \geq 1$, let $\mathcal{P}_\ell(d)$ be the space of degree-$\ell$ homogeneous polynomials with $d$ variables and real coefficients. Let $\mathcal{H}_\ell(d)$ denote the space of degree-$\ell$ harmonic polynomials in dimension $d$, i.e., homogeneous polynomial solutions of Laplace's equation:*

$$\mathcal{H}_\ell(d) := \{P \in \mathcal{P}_\ell(d) : \Delta P = 0\},$$

*where $\Delta = \frac{\partial^2}{\partial x_1^2} + \cdots + \frac{\partial^2}{\partial x_d^2}$ is the Laplace operator on $\mathbb{R}^d$. Finally, let $\mathcal{H}_\ell\left(\mathbb{S}^{d-1}\right)$ be the space of (real) Spherical Harmonics of order $\ell$ in dimension $d$, i.e. restrictions of harmonic polynomials in*

---

[2]Formally, $L^2\left(\mathbb{S}^{d-1}\right)$ is a space of equivalence classes of functions that differ at a set of points with measure 0. For notational simplicity, here and throughout we use $f$ to denote the specific representative of the equivalence class $f \in L^2\left(\mathbb{S}^{d-1}\right)$. In this way, we can consider the point-wise value $f(w)$ for every $w \in \mathbb{S}^{d-1}$.

$\mathcal{H}_\ell(d)$ to the sphere $\mathbb{S}^{d-1}$. The dimension of this space, $\alpha_{\ell,d} \equiv \dim\left(\mathcal{H}_\ell\left(\mathbb{S}^{d-1}\right)\right)$, is

$$\alpha_{0,d} = 1, \quad \alpha_{1,d} = d, \quad \alpha_{\ell,d} = \binom{d+\ell-1}{\ell} - \binom{d+\ell-3}{\ell-2} \quad \text{for } \ell \geq 2.$$

## 2.1 Gegenbauer Polynomials

The *Gegenbauer* (a.k.a. *ultraspherical*) *polynomial* of degree $\ell \geq 0$ in dimension $d \geq 2$ is given by

$$P_d^\ell(t) := \sum_{j=0}^{\lfloor \ell/2 \rfloor} c_j \cdot t^{\ell-2j} \cdot (1-t^2)^j, \tag{3}$$

where $c_0 = 1$ and $c_{j+1} = -\frac{(\ell-2j)(\ell-2j-1)}{2(j+1)(d-1+2j)} \cdot c_j$ for $j = 0, 1, \ldots, \lfloor \ell/2 \rfloor - 1$. These polynomials are orthogonal on the interval $[-1, 1]$ with respect to the measure $(1-t^2)^{\frac{d-3}{2}}$, i.e.,

$$\int_{-1}^{1} P_d^\ell(t) \cdot P_d^{\ell'}(t) \cdot (1-t^2)^{\frac{d-3}{2}} \, dt = \frac{\left|\mathbb{S}^{d-1}\right|}{\alpha_{\ell,d} \cdot \left|\mathbb{S}^{d-2}\right|} \cdot \mathbb{1}_{\{\ell=\ell'\}}. \tag{4}$$

**Zonal Harmonics.** The Gegenbauer polynomials naturally provide positive definite dot-product kernels on $\mathbb{S}^{d-1}$ known as *Zonal Harmonics*, which are closely related to the spherical harmonics. The following reproducing property of zonal harmonics plays a crucial role in our analysis.

**Lemma 1** (Reproducing Property of Zonal Harmonics). *Let $P_d^\ell(\cdot)$ be the Gengenbauer polynomial of degree $\ell$ in dimension $d$. For any $x, y \in \mathbb{S}^{d-1}$:*

$$P_d^\ell(\langle x, y \rangle) = \alpha_{\ell,d} \cdot \mathbb{E}_{w \sim \mathcal{U}(\mathbb{S}^{d-1})} \left[ P_d^\ell(\langle x, w \rangle) \, P_d^\ell(\langle y, w \rangle) \right],$$

*Furthermore, for any $\ell' \neq \ell$:*

$$\mathbb{E}_{w \sim \mathcal{U}(\mathbb{S}^{d-1})} \left[ P_d^\ell(\langle x, w \rangle) \cdot P_d^{\ell'}(\langle y, w \rangle) \right] = 0.$$

The proof of this and all subsequent results can be found in the appendix. The following useful fact, known as the addition theorem, connects Gegenbauer polynomials and spherical harmonics.

**Theorem 3** (Addition Theorem). *For every integer $\ell \geq 0$, if $\left\{ y_1^\ell, y_2^\ell, \ldots, y_{\alpha_{\ell,d}}^\ell \right\}$ is an orthonormal basis for $\mathcal{H}_\ell\left(\mathbb{S}^{d-1}\right)$, then for any $\sigma, w \in \mathbb{S}^{d-1}$ we have*

$$\frac{\alpha_{\ell,d}}{\left|\mathbb{S}^{d-1}\right|} \cdot P_d^\ell(\langle \sigma, w \rangle) = \sum_{j=1}^{\alpha_{\ell,d}} y_j^\ell(\sigma) \cdot y_j^\ell(w).$$

# 3 Reconstructing $L^2\left(\mathbb{S}^{d-1}\right)$ Functions via Spherical Harmonics

In this section we show how to approximate any function $f \in L^2\left(\mathbb{S}^{d-1}\right)$ by spherical harmonics using the optimal number of samples. We begin with the fact that spherical harmonics form a complete set of orthonormal functions and thus form an orthonormal basis for the Hilbert space of square-integrable functions on sphere $\mathbb{S}^{d-1}$. This is analogous to periodic functions, viewed as functions defined on the circle $\mathbb{S}^1$, which can be expressed as a linear combination of circular functions (sines and cosines) via the Fourier series.

**Lemma 2** (Direct Sum Decomposition of $L^2(\mathbb{S}^{d-1})$). *The family of spaces $\mathcal{H}_\ell\left(\mathbb{S}^{d-1}\right)$ yields a Hilbert space direct sum decomposition $L^2\left(\mathbb{S}^{d-1}\right) = \bigoplus_{\ell=0}^{\infty} \mathcal{H}_\ell\left(\mathbb{S}^{d-1}\right)$: the summands are closed and pairwise orthogonal, and every $f \in L^2\left(\mathbb{S}^{d-1}\right)$ is the sum of a converging series (in the sense of mean-square convergence with the $L^2$-norm defined in Eq. (2)),*

$$f = \sum_{\ell=0}^{\infty} f_\ell,$$

*where $f_\ell \in \mathcal{H}_\ell\left(\mathbb{S}^{d-1}\right)$ are uniquely determined functions. Furthermore, given any orthonormal basis $\left\{ y_1^\ell, y_2^\ell, \ldots, y_{\alpha_{\ell,d}}^\ell \right\}$ of $\mathcal{H}_\ell\left(\mathbb{S}^{d-1}\right)$ we have $f_\ell = \sum_{j=1}^{\alpha_{\ell,d}} \langle f, y_j^\ell \rangle_{\mathbb{S}^{d-1}} \cdot y_j^\ell$.*

The series expansion in Lemma 2 is the analog of the Fourier expansion of periodic functions, and is known as "*generalized Fourier series*" [Pen30] with respect to the Hilbert basis $\{y_j^\ell : j \in [\alpha_{\ell,d}], \ell \geq 0\}$. We remark that it is in general intractable to compute an orthogonal basis for the space of spherical harmonics [MNY06], which renders the generalized Fourier series expansion in Lemma 2 primarily existential. While finding the generalized Fourier expansion of a function $f \in L^2\left(\mathbb{S}^{d-1}\right)$ is computationally intractable, our goal is to answer the next fundamental question, which is about finding the projection of a function $f$ onto the space of spherical harmonics, i.e., the $f_\ell$'s in Lemma 2. Concretely, we seek to solve the following problem.

**Problem 2.** *For an integer $q \geq 0$ and a given function $f \in L^2\left(\mathbb{S}^{d-1}\right)$ whose decomposition over the Hilbert sum $\bigoplus_{\ell=0}^\infty \mathcal{H}_\ell\left(\mathbb{S}^{d-1}\right)$ is $f = \sum_{\ell=0}^\infty f_\ell$ as per Lemma 2, let us define the low-degree expansion of this function as $f^{(q)} := \sum_{\ell=0}^q f_\ell$. How efficiently can we learn $f^{(q)} \in \bigoplus_{\ell=0}^q \mathcal{H}_\ell\left(\mathbb{S}^{d-1}\right)$? More precisely, we want to find a set $\{w_1, w_2, \ldots, w_s\} \subseteq \mathbb{S}^{d-1}$ with minimal cardinality $s$ along with an efficient algorithm that given samples $\{f(w_i)\}_{i=1}^s$ can interpolate $f(\cdot)$ with a function $\tilde{f}^{(q)} \in \bigoplus_{\ell=0}^q \mathcal{H}_\ell\left(\mathbb{S}^{d-1}\right)$ such that:*

$$\left\| \tilde{f}^{(q)} - f^{(q)} \right\|_{\mathbb{S}^{d-1}}^2 \leq \varepsilon \cdot \left\| f^{(q)} - f \right\|_{\mathbb{S}^{d-1}}^2.$$

For ease of notation, we denote the Hilbert space of spherical harmonics of degree at most $q$ by $\mathcal{H}^{(q)}\left(\mathbb{S}^{d-1}\right) := \bigoplus_{\ell=0}^q \mathcal{H}_\ell\left(\mathbb{S}^{d-1}\right)$. To answer Problem 2 we exploit the close connection between the spherical harmonics and Gengenbauer polynomials, and in particular the fact that zonal harmonics are the reproducing kernels of the Hilbert spaces $\mathcal{H}_\ell\left(\mathbb{S}^{d-1}\right)$.

**Lemma 3** (A Reproducing Kernel for $\mathcal{H}_\ell\left(\mathbb{S}^{d-1}\right)$)**.** *For every $f \in L^2\left(\mathbb{S}^{d-1}\right)$, if $f = \sum_{\ell=0}^\infty f_\ell$ is the unique decomposition of $f$ over $\bigoplus_{\ell=0}^\infty \mathcal{H}_\ell\left(\mathbb{S}^{d-1}\right)$ as per Lemma 2, then $f_\ell$ is given by*

$$f_\ell(\sigma) = \alpha_{\ell,d} \cdot \mathop{\mathbb{E}}_{w \sim \mathcal{U}(\mathbb{S}^{d-1})} \left[ f(w) P_d^\ell\left(\langle \sigma, w\rangle\right) \right] \quad \text{for } \sigma \in \mathbb{S}^{d-1}.$$

Now we define a kernel operator, based on the low-degree Gegenbauer polynomials, which projects functions onto their low-degree spherical harmonic expansion.

**Definition 2** (Projection Operator onto $\mathcal{H}^{(q)}(\mathbb{S}^{d-1})$)**.** *For any integers $q \geq 0$ and $d \geq 2$, define the kernel operator $\mathcal{K}_d^{(q)} : L^2\left(\mathbb{S}^{d-1}\right) \to L^2\left(\mathbb{S}^{d-1}\right)$ as follows: for $f \in L^2\left(\mathbb{S}^{d-1}\right)$ and $\sigma \in \mathbb{S}^{d-1}$,*

$$\left[\mathcal{K}_d^{(q)} f\right](\sigma) := \sum_{\ell=0}^q \frac{\alpha_{\ell,d}}{|\mathbb{S}^{d-1}|} \left\langle f, P_d^\ell\left(\langle \sigma, \cdot\rangle\right)\right\rangle_{\mathbb{S}^{d-1}} = \sum_{\ell=0}^q \alpha_{\ell,d} \cdot \mathop{\mathbb{E}}_{w \sim \mathcal{U}(\mathbb{S}^{d-1})} \left[ f(w) P_d^\ell\left(\langle \sigma, w\rangle\right) \right]. \quad (5)$$

*This is an integral operator with kernel function $k_{q,d}(\sigma, w) := \sum_{\ell=0}^q \frac{\alpha_{\ell,d}}{|\mathbb{S}^{d-1}|} \cdot P_d^\ell\left(\langle \sigma, w\rangle\right)$.*

Note that the operator $\mathcal{K}_d^{(q)}$ is self-adjoint and positive semi-definite. Moreover, using the reproducing property of this kernel we can establish that $\mathcal{K}_d^{(q)}$ is a projection operator.

**Claim 1.** *The operator $\mathcal{K}_d^{(q)}$ defined in Definition 2 satisfies the property $\left(\mathcal{K}_d^{(q)}\right)^2 = \mathcal{K}_d^{(q)}$.*

Furthermore, by the addition theorem (Theorem 3), $\mathcal{K}_d^{(q)}$ is trace-class (i.e., the trace is finite and independent of the choice of basis) because:

$$\text{trace}\left(\mathcal{K}_d^{(q)}\right) = \int_{\mathbb{S}^{d-1}} k_{q,d}(w, w) \, dw = \sum_{\ell=0}^q \frac{\alpha_{\ell,d}}{|\mathbb{S}^{d-1}|} \cdot \int_{\mathbb{S}^{d-1}} P_d^\ell\left(\langle w, w\rangle\right) \, dw$$

$$= \sum_{\ell=0}^q \alpha_{\ell,d} = \binom{d+q-1}{q} + \binom{d+q-2}{q-1} - 1. \quad (6)$$

By combining Theorem 3 and Lemma 2, and using the definition of the projection operator $\mathcal{K}_d^{(q)}$, it follows that for any function $f \in L^2\left(\mathbb{S}^{d-1}\right)$ with Hilbert sum decomposition $f = \sum_{\ell=0}^\infty f_\ell$, the low-degree component $f^{(q)} = \sum_{\ell=0}^q f_\ell \in \mathcal{H}^{(q)}\left(\mathbb{S}^{d-1}\right)$ can be computed as $f^{(q)} = \mathcal{K}_d^{(q)} f$. Equivalently,

in order to learn $f^{(q)}$, it suffices to solve the following least-squares regression problem,

$$\min_{g \in L^2(\mathbb{S}^{d-1})} \left\| \mathcal{K}_d^{(q)} g - f \right\|_{\mathbb{S}^{d-1}}^2. \tag{7}$$

If $g^*$ is an optimal solution to the above regression problem then $f^{(q)} = \mathcal{K}_d^{(q)} g^*$. In the next claim we show that solving the least squares problem in Eq. (7), even to a coarse approximation, is sufficient to solve our interpolation problem (i.e., Problem 2):

**Claim 2.** *For any $f \in L^2\left(\mathbb{S}^{d-1}\right)$, any integer $q \geq 0$, and any $C \geq 1$, if $\tilde{g} \in L^2\left(\mathbb{S}^{d-1}\right)$ satisfies,*

$$\left\| \mathcal{K}_d^{(q)} \tilde{g} - f \right\|_{\mathbb{S}^{d-1}}^2 \leq C \cdot \min_{g \in L^2(\mathbb{S}^{d-1})} \left\| \mathcal{K}_d^{(q)} g - f \right\|_{\mathbb{S}^{d-1}}^2,$$

*and if we let $f^{(q)} := \mathcal{K}_d^{(q)} f$, where $\mathcal{K}_d^{(q)}$ is defined as per Definition 2, then the following holds*

$$\left\| \mathcal{K}_d^{(q)} \tilde{g} - f^{(q)} \right\|_{\mathbb{S}^{d-1}}^2 \leq (C-1) \cdot \left\| f^{(q)} - f \right\|_{\mathbb{S}^{d-1}}^2.$$

Claim 2 shows that solving the regression problem in Eq. (7) approximately provides a solution to our spherical harmonics interpolation problem (Problem 2). But how can we solve this least-squares problem efficiently? Not only does the problem involve a possibly infinite dimensional parameter vector $g$, but the objective function also involves the continuous domain on the surface of $\mathbb{S}^{d-1}$.

## 3.1 Randomized Discretization via Leverage Function Sampling

We solve the continuous regression in Eq. (7) by randomly discretizing the sphere $\mathbb{S}^{d-1}$, thereby reducing our problem to a regression on a finite set of points $w_1, w_2, \ldots, w_s \in \mathbb{S}^{d-1}$. In particular, we propose to sample points on $\mathbb{S}^{d-1}$ with probability proportional to the so-called *leverage function*, a specific distribution that has been widely applied in randomized algorithms for linear algebra problems on discrete matrices [LMP13]. We start with the definition of the leverage function for compact operators such as $\mathcal{K}_d^{(q)}$:

**Definition 3** (Leverage Function). *For integers $q \geq 0$ and $d > 0$, we define the leverage function of the operator $\mathcal{K}_d^{(q)}$ (see Definition 2) for every $w \in \mathbb{S}^{d-1}$ as follows,*

$$\tau_q(w) := \max_{g \in L^2(\mathbb{S}^{d-1})} \left\| \mathcal{K}_d^{(q)} g \right\|_{\mathbb{S}^{d-1}}^{-2} \cdot \left| \left[ \mathcal{K}_d^{(q)} g \right] (w) \right|^2. \tag{8}$$

Intuitively, $\tau_q(w)$ is an upper bound of how much a function that is spanned by the eigenfunctions of the operator $\mathcal{K}_d^{(q)}$ can "blow up" at $w$. The larger the leverage function $\tau_q(w)$ implies the higher the probability we will be required to sample $w$. This ensures that our sample points well reflect any possibly significant components, or "spikes", of the function. Ultimately, the integral $\int_{\mathbb{S}^{d-1}} \tau_q(w)\, dw$ determines how many samples we require to solve the regression problem Eq. (7) to a given accuracy. It is a known fact that the leverage function integrates to the rank of the operator $\mathcal{K}_d^{(q)}$ (which is equal to the dimensionality of the Hilbert space $\mathcal{H}^{(q)}(\mathbb{S}^{d-1})$). This ultimately allows us to achieve a $\widetilde{\mathcal{O}}(\sum_{\ell=0}^q \alpha_{\ell,d})$ sample complexity bound for solving Problem 2. To compute the leverage function, we make use of the following useful alternative characterization of the leverage function.

**Lemma 4** (Min Characterization of the Leverage Function). *For any $w \in \mathbb{S}^{d-1}$, let $\tau_q(w)$ be the leverage function (Definition 3) and define $\phi_w \in L^2(\mathbb{S}^{d-1})$ by $\phi_w(\sigma) \equiv \sum_{\ell=0}^q \frac{\alpha_{\ell,d}}{|\mathbb{S}^{d-1}|} P_d^\ell(\langle \sigma, w \rangle)$. We have the following minimization characterization of the leverage function:*

$$\tau_q(w) = \left\{ \min_{g \in L^2(\mathbb{S}^{d-1})} \|g\|_{\mathbb{S}^{d-1}}^2, \quad s.t. \ \mathcal{K}_d^{(q)} g = \phi_w \right\}. \tag{9}$$

Using the min and max characterizations of the leverage function we can find upper and lower bounds on this function. Surprisingly, in this case the upper and lower bounds match, so we actually have an exact value for the leverage function.

**Lemma 5** (Leverage Function is Constant). *The leverage function given in Definition 3 is equal to $\tau_q(w) = \sum_{\ell=0}^q \frac{\alpha_{\ell,d}}{|\mathbb{S}^{d-1}|}$ for every $w \in \mathbb{S}^{d-1}$.*

---

**Algorithm 1** Efficient Spherical Harmonic Expansion

---

1: **Input:** accuracy parameter $\varepsilon > 0$, integer $q \geq 0$
2: Set $s = c \cdot (\beta_{q,d} \log \beta_{q,d} + \beta_{q,d}/\varepsilon)$ for sufficiently large fixed constant $c$
3: Sample i.i.d. random points $w_1, w_2, \ldots, w_s$ from $\mathcal{U}(\mathbb{S}^{d-1})$
4: Compute $\boldsymbol{K} \in \mathbb{R}^{s \times s}$ with $\boldsymbol{K}_{i,j} = \sum_{\ell=0}^{q} \frac{\alpha_{\ell,d}}{s \cdot |\mathbb{S}^{d-1}|} \cdot P_d^\ell (\langle w_i, w_j \rangle)$ for $i, j \in [s]$
5: Compute $\boldsymbol{f} \in \mathbb{R}^s$ with $\boldsymbol{f}_j = \frac{1}{\sqrt{s}} \cdot f(w_j)$ for $j \in [s]$
6: Solve the regression by computing $\boldsymbol{z} = \boldsymbol{K}^\dagger \boldsymbol{f}$
7: **Return:** $y \in \mathcal{H}^{(q)}(\mathbb{S}^{d-1})$ with $y(\sigma) := \sum_{\ell=0}^{q} \frac{\alpha_{\ell,d}}{\sqrt{s} \cdot |\mathbb{S}^{d-1}|} \cdot \sum_{j=1}^{s} \boldsymbol{z}_j \cdot P_d^\ell (\langle w_j, \sigma \rangle)$ for $\sigma \in \mathbb{S}^{d-1}$

---

We prove this lemma in Appendix C. The integral of the leverage function, which determines the total samples needed to solve our least-squares regression, is therefore equal to the dimensionality of the Hilbert space $\mathcal{H}^{(q)}(\mathbb{S}^{d-1})$.

**Corollary 1.** *The leverage function defined in Definition 3 integrates to the dimensionality of the Hilbert space $\mathcal{H}^{(q)}(\mathbb{S}^{d-1})$, which we denote by $\beta_{q,d}$, i.e.,*

$$\int_{\mathbb{S}^{d-1}} \tau_q(w) \, dw = \dim \left( \mathcal{H}^{(q)}(\mathbb{S}^{d-1}) \right) = \sum_{\ell=0}^{q} \alpha_{\ell,d} \equiv \beta_{q,d}.$$

We now show that the leverage function can be used to randomly sample the points on the unit sphere to discretize the regression problem in Eq. (7) and solve it approximately.

**Theorem 4** (Approximate Regression via Leverage Function Sampling). *For any $\varepsilon > 0$, let $s = c \cdot \left( \beta_{q,d} \log \beta_{q,d} + \frac{\beta_{q,d}}{\varepsilon} \right)$, for sufficiently large fixed constant $c$, and let $x_1, x_2, \ldots, x_s$ be i.i.d. uniform samples on $\mathbb{S}^{d-1}$. Define the quasi-matrix $\boldsymbol{P} : \mathbb{R}^s \to L^2(\mathbb{S}^{d-1})$ as follows, for every $v \in \mathbb{R}^d$:*

$$[\boldsymbol{P} v](\sigma) := \sum_{\ell=0}^{q} \frac{\alpha_{\ell,d}}{\sqrt{s} \cdot |\mathbb{S}^{d-1}|} \cdot \sum_{j=1}^{s} v_j \cdot P_d^\ell (\langle x_j, \sigma \rangle) \quad \text{for } \sigma \in \mathbb{S}^{d-1}.$$

*Also let $\boldsymbol{f} \in \mathbb{R}^s$ be a vector with $\boldsymbol{f}_j := \frac{1}{\sqrt{s}} \cdot f(x_j)$ for $j = 1, 2, \ldots, s$ and let $\boldsymbol{P}^*$ be the adjoint of $\boldsymbol{P}$. If $\tilde{g}$ is an optimal solution to the least-squares problem $\tilde{g} \in \arg \min_{g \in L^2(\mathbb{S}^{d-1})} \|\boldsymbol{P}^* g - \boldsymbol{f}\|_2^2$, then with probability at least $1 - 10^{-4}$ the following holds,*

$$\left\| \mathcal{K}_d^{(q)} \tilde{g} - f \right\|_{\mathbb{S}^{d-1}}^2 \leq (1 + \varepsilon) \cdot \min_{g \in L^2(\mathbb{S}^{d-1})} \left\| \mathcal{K}_d^{(q)} g - f \right\|_{\mathbb{S}^{d-1}}^2.$$

We prove Theorem 4 in Appendix C. This theorem shows that the function $\tilde{g}$ obtained from solving the discretized regression problem provides an approximate solution to Eq. (7).

## 3.2 Efficient Solution for the Discretized Least-Squares Problem

In this section, we demonstrate how to apply Theorem 4 algorithmically to approximately solve the regression problem of Eq. (7). To achieve this, we leverage the *kernel trick*, following a similar approach to previous works such as [AKM+19, CP19], which allows us to efficiently address the randomly discretized least squares problem as detailed in Algorithm 1. The associated guarantee for this approach is provided in Theorem 5.

**Theorem 5** (Efficient Spherical Harmonic Interpolation). *Algorithm 1 returns a function $y \in \mathcal{H}^{(q)}(\mathbb{S}^{d-1})$ such that, with probability at least $1 - 10^{-4}$:*

$$\left\| y - f^{(q)} \right\|_{\mathbb{S}^{d-1}}^2 \leq \varepsilon \cdot \left\| f^{(q)} - f \right\|_{\mathbb{S}^{d-1}}^2, \quad \text{where } f^{(q)} := \mathcal{K}_d^{(q)} f.$$

*Suppose we can compute the Gegenbauer polynomial $P_d^\ell(t)$ at every point $t \in [-1, 1]$ in constant time. Then Algorithm 1 queries the function $f$ at $s = \mathcal{O} \left( \beta_{q,d} \log \beta_{q,d} + \frac{\beta_{q,d}}{\varepsilon} \right)$ points on the sphere $\mathbb{S}^{d-1}$ and runs in $\mathcal{O}(s^2 \cdot d + s^\omega)$ time. This algorithm evaluates $y(\sigma)$ in $\mathcal{O}(d \cdot s)$ time for any $\sigma \in \mathbb{S}^{d-1}$.*

We provide the proof of this theorem in Appendix D.

**Remark 1** (Noise Robustness). *In Theorem 5, our method's robustness is demonstrated under a noise model where the function $f$ is not strictly a low-degree spherical harmonic and may include high-degree components. In this scenario, the higher-degree components are considered as noise added to the input function.*

*However, our algorithm is robust against alternative noise models, particularly additive i.i.d. Normal noise that corrupts the function evaluations $\boldsymbol{f}$ in Algorithm 1 with iid Normal noise. More precisely, suppose that we observe samples from the function $f^{(q)}$ contaminated by Gaussian noise, i.e., $f(w_j) = f^{(q)}(w_j) + n_j$ in Algorithm 1 for i.i.d. $n_1, n_2, \ldots n_s \sim \mathcal{N}(0,1)$. The expected value of perturbation's norm in the output $y$ of our algorithm caused by this noise is:*

$$\mathbb{E}\left[\left\|y - f^{(q)}\right\|_{\mathbb{S}^{d-1}}^2\right] = (1/s) \cdot \operatorname{trace}\left(\boldsymbol{K}^\dagger \boldsymbol{K} \boldsymbol{K}^\dagger\right).$$

*By Markov's inequality, with $0.99$ probability $\left\|y - f^{(q)}\right\|_{\mathbb{S}^{d-1}}^2 = O(1/s) \cdot \operatorname{trace}\left(\boldsymbol{K}^\dagger \boldsymbol{K} \boldsymbol{K}^\dagger\right)$. If we let $\boldsymbol{P}$ be the operator defined in Theorem 4, then we can see that $\boldsymbol{K} = \boldsymbol{P}^* \boldsymbol{P}$. Using the properties of the trace, one can see that $\operatorname{trace}\left(\boldsymbol{K}^\dagger \boldsymbol{K} \boldsymbol{K}^\dagger\right) = 1/\lambda_1 + 1/\lambda_2 + \ldots$, where $\lambda_i$'s are the singular values of the operator $\boldsymbol{P}\boldsymbol{P}^*$. By matrix Chernoff inequalities one can show that all singular values of the operator $\boldsymbol{P}\boldsymbol{P}^*$ closely approximate the singular values of the projection operator $\mathcal{K}_d^{(q)}$ up to a constant factor. So, we have $\operatorname{trace}\left(\boldsymbol{K}^\dagger \boldsymbol{K} \boldsymbol{K}^\dagger\right) = O(\operatorname{rank}(\mathcal{K}_d^{(q)})) = O(\beta_{q,d})$. Thus, because $s \geq \Omega(\beta/\varepsilon)$ and using union bound, with $0.98$ probability:*

$$\left\|y - f^{(q)}\right\|_{\mathbb{S}^{d-1}}^2 \leq \varepsilon.$$

## 4 Lower Bound on The Number of Required Observations

We conclude by showing that the dimensionality of the Hilbert space $\mathcal{H}^{(q)}(\mathbb{S}^{d-1})$ tightly characterizes the sample complexity of Problem 2. Thus, our Theorem 5 is optimal up to a logarithmic factor. Intuitively, there are $\beta_{q,d}$ degrees of freedom for specifying a spherical harmonic. Consequently, any deterministic algorithm attempting to reconstruct such polynomials would need at least $\beta_{q,d}$ samples. We aim to demonstrate that even a "randomized" algorithm, which succeeds with only a constant probability, must still gather $\beta_{q,d}$ samples. This complements our upper bound, which is established using a randomized algorithm. The crucial fact we use for proving the lower bound is that all (non-zero) eigenvalues of the operator $\mathcal{K}_d^{(q)}$ are equal to one. This fact follows from the addition theorem presented in Theorem 3, i.e., if $\left\{y_1^\ell, y_2^\ell, \ldots, y_{\alpha_{\ell,d}}^\ell\right\}$ is an orthonormal basis of $\mathcal{H}_\ell\left(\mathbb{S}^{d-1}\right)$, then for any function $f \in L^2\left(\mathbb{S}^{d-1}\right)$,

$$\left[\mathcal{K}_d^{(q)} f\right](\sigma) = \sum_{\ell=0}^q \alpha_{\ell,d} \cdot \mathbb{E}_{w \sim \mathcal{U}(\mathbb{S}^{d-1})}\left[P_d^\ell\left(\langle \sigma, w \rangle\right) \cdot f(w)\right] = \sum_{\ell=0}^q \sum_{j=1}^{\alpha_{\ell,d}} \langle y_j^\ell, f \rangle_{\mathbb{S}^{d-1}} \cdot y_j^\ell(\sigma). \quad (10)$$

**Theorem 6** (Lower Bound). *Consider an error parameter $\varepsilon > 0$, and any (possibly randomized) algorithm that solves Problem 2 with probability greater than $1/10$ for any input function $f$ and makes at most $r$ (possibly adaptive) queries on any input. Then $r \geq \beta_{q,d}$.*

We describe a distribution on input function $f$ on which any deterministic algorithm that takes $r < \beta_{q,d}$ samples fails with probability $\geq 9/10$. The theorem then follows by Yao's principle.

**Hard Input Distribution.** For integer $\ell \leq q$, consider an orthonormal basis of $\mathcal{H}_\ell\left(\mathbb{S}^{d-1}\right)$ and denote it by $\left\{y_1^\ell, y_2^\ell, \ldots, y_{\alpha_{\ell,d}}^\ell\right\}$. Let $\boldsymbol{Y}_\ell : \mathbb{R}^{\alpha_{\ell,d}} \to \mathcal{H}_\ell\left(\mathbb{S}^{d-1}\right)$ be the quasi-matrix with $y_j^\ell$ as its $j^{th}$ column, i.e., $[\boldsymbol{Y}_\ell \cdot u](\sigma) := \sum_{j=1}^{\alpha_{\ell,d}} u_j \cdot y_j^\ell(\sigma)$ for any $u \in \mathbb{R}^{\alpha_{\ell,d}}$ and $\sigma \in \mathbb{S}^{d-1}$. Let $v^{(0)} \in \mathbb{R}^{\alpha_{0,d}}, v^{(1)} \in \mathbb{R}^{\alpha_{1,d}}, \ldots, v^{(q)} \in \mathbb{R}^{\alpha_{q,d}}$ be independent random vectors with i.i.d. Gaussian entries: $v_j^{(\ell)} \sim \mathcal{N}(0,1)$. The random input is defined to be $f := \sum_{\ell=0}^q \boldsymbol{Y}_\ell \cdot v^{(\ell)}$. In other words, $f = \sum_{\ell=0}^q \boldsymbol{Y}_\ell \cdot v^{(\ell)}$ is a random linear combination of the eigenfunctions of $\mathcal{K}_d^{(q)}$. We prove that accurate

reconstruction of $f$ drawn from the aforementioned distribution yields accurate reconstruction of random vectors $v^{(0)}, v^{(1)}, \ldots, v^{(q)}$. Since each $v^{(\ell)}$ is $\alpha_{\ell,d}$-dimensional, this reconstruction requires $\Omega(\sum_{\ell=0}^{q} \alpha_{\ell,d}) = \Omega(\beta_{q,d})$ samples, giving us a lower bound for accurate reconstruction of $f$.

First we show that finding an $\tilde{f}^{(q)}$ satisfying the condition of Problem 2 is at least as hard as accurately finding all vectors $v^{(0)}, v^{(1)}, \ldots, v^{(q)}$. The following lemma is proved in Appendix E.

**Lemma 6.** *If a deterministic algorithm solves Problem 2 with probability at least $1/10$ over our random input distribution $f = \sum_{\ell=0}^{q} \boldsymbol{Y}_\ell \cdot v^{(\ell)}$, then with probability at least $1/10$, the output of the algorithm $\tilde{f}^{(q)}$ satisfies $\boldsymbol{Y}_\ell^* \tilde{f}^{(q)} = v^{(\ell)}$ for all integers $\ell \leq q$.*

Finally, we complete the proof of Theorem 6 by arguing that if $\tilde{f}^{(q)}$ is formed using less than $\beta_{q,d}$ queries from $f$, then $\sum_{\ell=0}^{q} \left\| \boldsymbol{Y}_\ell^* \tilde{f}^{(q)} - v^{(\ell)} \right\|_2^2 > 0$ with good probability, thus the bound of Lemma 6 cannot hold and $\tilde{f}^{(q)}$ cannot be a solution to Problem 2. Assume for sake of contradiction that there is a deterministic algorithm which solves Problem 2 with probability $\geq 1/10$ over the random input $f = \sum_{\ell=0}^{q} \boldsymbol{Y}_\ell \cdot v^{(\ell)}$ that makes $r = \beta_{q,d} - 1$ queries on any input (we can modify an algorithm that makes fewer queries to make exactly $\beta_{\ell,d} - 1$ queries). For every $\sigma \in \mathbb{S}^{d-1}$ and integer $\ell \leq q$ define $u_\sigma^\ell := \left[ y_1^\ell(\sigma), y_2^\ell(\sigma), \ldots, y_{\alpha_{\ell,d}}^\ell(\sigma) \right]$. Also define $\boldsymbol{u}_\sigma := \left[ u_\sigma^0, u_\sigma^1, \ldots, u_\sigma^q \right] \in \mathbb{R}^{\beta_{q,d}}$ and $\boldsymbol{v} \in \mathbb{R}^{\beta_{q,d}}$ as $\boldsymbol{v} := \left( v^{(0)}, v^{(1)}, \ldots, v^{(q)} \right)$. Additionally, define the quasi-matrix $\boldsymbol{Y} := [\boldsymbol{Y}_0, \ldots, \boldsymbol{Y}_q]$.

Using the above notations and the definition of the hard input instance $f$, each query to $f$ is in fact a query to the random vector $\boldsymbol{v}$ in the form of $f(\sigma) = \langle \boldsymbol{u}_\sigma, \boldsymbol{v} \rangle$. Now consider a deterministic function $Q$, that is given input $\boldsymbol{V} \in \mathbb{R}^{i \times \beta_{q,d}}$ (for any positive integer $i$) and outputs $Q(\boldsymbol{V}) \in \mathbb{R}^{\beta_{q,d} \times \beta_{q,d}}$ such that $Q(\boldsymbol{V})$ has orthonormal rows with the first $i$ rows spanning the $i$ rows of $\boldsymbol{V}$. If $\sigma_1, \sigma_2, \ldots, \sigma_r \in \mathbb{S}^{d-1}$ denote the points where our algorithm queries the input $f$, for any integer $i \in [r]$, let $\boldsymbol{Q}^i$ be an orthonormal matrix whose first $i$ rows span the first $i$ queries of the algorithm, i.e., $\boldsymbol{Q}^i := Q \left( [\boldsymbol{u}_{\sigma_1}, \boldsymbol{u}_{\sigma_2}, \ldots, \boldsymbol{u}_{\sigma_i}]^\top \right)$. Since the algorithm is deterministic, $\boldsymbol{Q}^i$ is a deterministic function of input $\boldsymbol{v}$. The following claim is proved in [AKM+19]:

**Claim 3** (Claim 23 of [AKM+19]). *Conditioned on the queries $f(\sigma_1), f(\sigma_2), \ldots, f(\sigma_r)$ for $r < \beta_{q,d}$, the variable $[\boldsymbol{Q}^r \cdot \boldsymbol{v}](\beta_{q,d})$ is distributed as $\mathcal{N}(0,1)$.*

Now using Claim 3 we can write,

$$\Pr_{\boldsymbol{v}} \left[ \sum_{\ell=0}^{q} \left\| v^{(\ell)} - \boldsymbol{Y}_\ell^* \tilde{f}^{(q)} \right\|_2^2 = 0 \right] = \Pr_{\boldsymbol{v}} \left[ \boldsymbol{Q}^r \boldsymbol{v} = \boldsymbol{Q}^r \boldsymbol{Y}^* \tilde{f}^{(q)} \right] \leq \Pr_{\boldsymbol{v}} \left[ \left[ \boldsymbol{Q}^r \boldsymbol{v} \right]_{\beta_{q,d}} = \left[ \boldsymbol{Q}^r \boldsymbol{Y}^* \tilde{f}^{(q)} \right]_{\beta_{q,d}} \right]$$

$$= \mathbb{E} \left[ \Pr_{\boldsymbol{v}} \left[ \left[ \boldsymbol{Q}^r \boldsymbol{v} \right]_{\beta_{q,d}} = \left[ \boldsymbol{Q}^r \boldsymbol{Y}^* \tilde{f}^{(q)} \right]_{\beta_{q,d}} \, \middle| \, f(\sigma_1), \ldots, f(\sigma_r) \right] \right],$$

where the expectation in the last line is taken over the randomness of $f(\sigma_1), \ldots, f(\sigma_r)$. Conditioned on $f(\sigma_1), \ldots, f(\sigma_r)$, $\left[ \boldsymbol{Q}^r \boldsymbol{Y}^* \tilde{f}^{(q)} \right] (\beta_{q,d})$ is a fixed quantity because the algorithm determines $\tilde{f}^{(q)}$ given the knowledge of the queries $f(\sigma_1), \ldots, f(\sigma_r)$. Furthermore, by Claim 3, $[\boldsymbol{Q}^r \cdot \boldsymbol{v}](\beta_{q,d})$ is a random variable distributed as $\mathcal{N}(0,1)$, conditioned on $f(\sigma_1), \ldots, f(\sigma_r)$. This implies that,

$$\Pr \left[ \left[ \boldsymbol{Q}^r \cdot \boldsymbol{v} \right] (\beta_{q,d}) = \left[ \boldsymbol{Q}^r \boldsymbol{Y}^* \tilde{f}^{(q)} \right] (\beta_{q,d}) \, \middle| \, f(\sigma_1), \ldots, f(\sigma_r) \right] = 0.$$

Thus, $\Pr \left[ \sum_{\ell=0}^{q} \left\| v^{(\ell)} - \boldsymbol{Y}_\ell^* \tilde{f}^{(q)} \right\|_2^2 = 0 \right] = \mathbb{E}_{f(\sigma_1), \ldots, f(\sigma_r)}[0] = 0$. However, we have assumed that this algorithm solves Problem 2 with probability at least $1/10$, and hence, by Lemma 6, $\Pr \left[ \sum_{\ell=0}^{q} \| v^{(\ell)} - \boldsymbol{Y}_\ell^* \tilde{f}^{(q)} \|_2^2 = 0 \right] \geq 1/10$. This is a contradiction, yielding Theorem 6.

## 5 Numerical Evaluation

**Noise-free Setting.** For a fixed $q$, we generate a random function $f(\sigma) = \sum_{\ell=0}^{q} c_\ell P_d^\ell(\langle \sigma, v \rangle)$ where $v \sim \mathcal{U}(\mathbb{S}^{d-1})$ and $c_\ell$'s are i.i.d. samples from $\mathcal{N}(0,1)$. Then, $f$ is recovered by running Algorithm 1 with $s$ random evaluations of $f$ on $\mathbb{S}^{d-1}$. Note that $\|\mathcal{K}_d^{(q)} f - f\|_{\mathbb{S}^{d-1}} = 0$ since $f \in \mathcal{H}^{(q)}(\mathbb{S}^{d-1})$, thus,

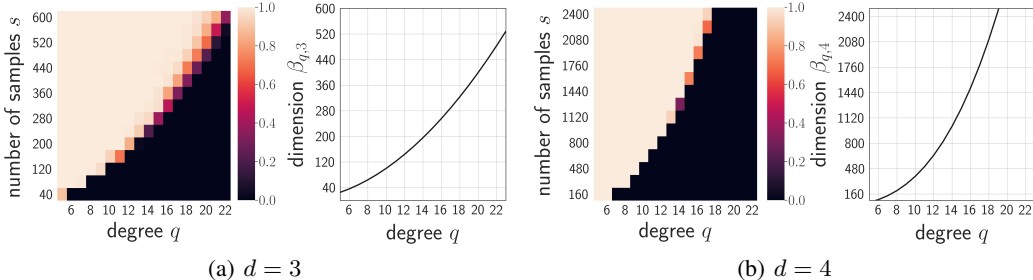

(a) $d = 3$          (b) $d = 4$

Figure 1: (Left) Empirical success probabilities of Algorithm 1 varying the number of samples $s$ and the degree of spherical harmonic expansion $q$. (Right) The dimension $\beta_{q,d}$ of the Hilbert space $\mathcal{H}^{(q)}(\mathbb{S}^{d-1})$ as a function of $q$ when (a) $d = 3$ and (b) $d = 4$, respectively.

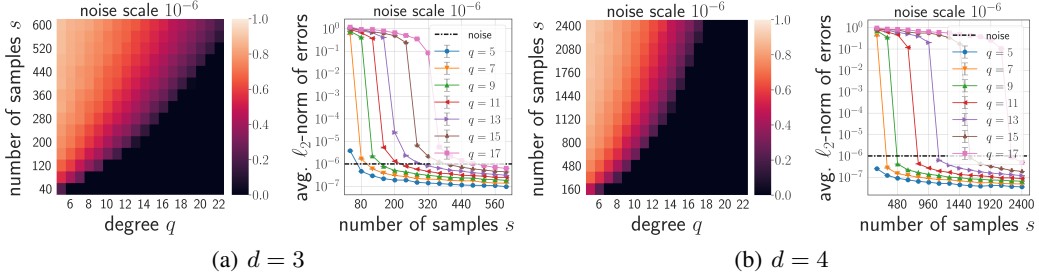

(a) $d = 3$          (b) $d = 4$

Figure 2: (Left) Empirical success probabilities of Algorithm 1 in presence of additive noise, where "success" means the error's energy is below the noise level $\|\tilde{f}^{(q)} - f\|_{\mathbb{S}^{d-1}} \leq \|n\|_{\mathbb{S}^{d-1}}$. (Right) The error's norm $\|\tilde{f}^{(q)} - f\|_{\mathbb{S}^{d-1}}$ as a function of $q$ when (a) $d = 3$ and (b) $d = 4$, respectively.

as shown in Theorem 4, Algorithm 1 can recover $f$ "exactly" using $s = \mathcal{O}(\beta_{q,d} \log \beta_{q,d})$ evaluations, where $\beta_{q,d}$ is the dimension of the Hilbert space $\mathcal{H}^{(q)}(\mathbb{S}^{d-1})$. We predict $f$'s value on a random test set on $\mathbb{S}^{d-1}$ and consider the algorithm fails if the testing error is greater than $10^{-12}$. We count the number of failures among 100 independent random trials with different choices of $d \in \{3, 4\}$, $q \in \{5, \ldots, 22\}$, and $s \in \{40, \ldots, 2400\}$. The empirical success probabilities for $d = 3$ and 4 are reported in Fig. 1(a) and Fig. 1(b), respectively. Fig. 1 illustrates that the success probabilities of our algorithm sharply transition to 1 as soon as the number of samples approaches $s \approx \beta_{q,d}$ for a wide range of $q$ and both $d = 3, 4$. These experimental results complement our Theorem 4 along with the lower bound analysis in Section 4 and empirically verify the performance of our algorithm.

**Noisy Setting.** We repeated our experiments in the presence of an additive noise which is a linear combination of random spherical harmonics of degrees $q + 1$ to $2q$. More precisely, we let the noise be $n(\sigma) = \sum_{\ell=q+1}^{2q} c_\ell P_d^\ell(\langle v, \sigma \rangle) \in \mathcal{H}^{(2q)}(\mathbb{S}^{d-1}) \setminus \mathcal{H}^{(q)}(\mathbb{S}^{d-1})$ for $c_\ell$'s that are i.i.d. samples from $\mathcal{N}(0, 1)$. We then re-scale the noise to have norm $\|n\|_{\mathbb{S}^{d-1}} = 10^{-6}$. Furthermore, the function $f$ is defined as before, and $f$ is recovered by Algorithm 1 with $s$ random evaluations of $f + n$ on $\mathbb{S}^{d-1}$. The heat-maps in Fig. 2 are generated by considering an instance of our algorithm as a "success" if the error's energy is below the noise level, $\left\|\tilde{f}^{(q)} - f\right\|_{\mathbb{S}^{d-1}} \leq \|n\|_{\mathbb{S}^{d-1}} = 10^{-6}$. The success probability transitions less sharply than the noiseless setting but the shift of probabilities starts at $\beta_{s,q}$ samples.

# 6 Conclusion

We studied the problem of robustly recovering spherical harmonic expansion of a function defined on the sphere. The number of function evaluations needed to recover its degree-$q$ expansion is the dimension of spherical harmonics of degree at most $q$, up to a logarithmic factor. We develop a simple yet efficient kernel regression-based algorithm to recover degree-$q$ expansion of the function by only evaluating the function on uniformly sampled points on the sphere. Unlike the prior results on fast spherical harmonic transform, our algorithm works efficiently using a nearly optimal number of samples in any dimension. We believe our findings would appeal to the readership of the community.

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

# A  Properties of Gegenbauer Polynomials and Spherical Harmonics

In this section we prove the basic properties of the Gegenbauer Polynomials as well as the Spherical Harmonics and establish the connection between the two. We start by the direct sum decomposition of the Hilbert space $L^2(\mathbb{S}^{d-1})$ in terms of the spherical harmonics,

**Lemma 2** (Direct Sum Decomposition of $L^2(\mathbb{S}^{d-1})$). *The family of spaces $\mathcal{H}_\ell\left(\mathbb{S}^{d-1}\right)$ yields a Hilbert space direct sum decomposition $L^2\left(\mathbb{S}^{d-1}\right) = \bigoplus_{\ell=0}^\infty \mathcal{H}_\ell\left(\mathbb{S}^{d-1}\right)$: the summands are closed and pairwise orthogonal, and every $f \in L^2\left(\mathbb{S}^{d-1}\right)$ is the sum of a converging series (in the sense of mean-square convergence with the $L^2$-norm defined in Eq. (2)),*

$$f = \sum_{\ell=0}^\infty f_\ell,$$

*where $f_\ell \in \mathcal{H}_\ell\left(\mathbb{S}^{d-1}\right)$ are uniquely determined functions. Furthermore, given any orthonormal basis $\left\{y_1^\ell, y_2^\ell, \ldots, y_{\alpha_{\ell,d}}^\ell\right\}$ of $\mathcal{H}_\ell\left(\mathbb{S}^{d-1}\right)$ we have $f_\ell = \sum_{j=1}^{\alpha_{\ell,d}} \langle f, y_j^\ell\rangle_{\mathbb{S}^{d-1}} \cdot y_j^\ell$.*

*Proof.* This is in fact a standard result. For example, see [Lan12] for a proof.

$\square$

Now we show that the Gegenbauer polynomials and spherical harmonics are related through the so called the addition theorem,

**Theorem 3** (Addition Theorem). *For every integer $\ell \geq 0$, if $\left\{y_1^\ell, y_2^\ell, \ldots, y_{\alpha_{\ell,d}}^\ell\right\}$ is an orthonormal basis for $\mathcal{H}_\ell\left(\mathbb{S}^{d-1}\right)$, then for any $\sigma, w \in \mathbb{S}^{d-1}$ we have*

$$\frac{\alpha_{\ell,d}}{|\mathbb{S}^{d-1}|} \cdot P_d^\ell\left(\langle\sigma, w\rangle\right) = \sum_{j=1}^{\alpha_{\ell,d}} y_j^\ell(\sigma) \cdot y_j^\ell(w).$$

*Proof.* The result can be proven analytically, using the properties of the Poisson kernel in the unit ball. This is classic and the proof can be found in [AH12, Theorem 2.9].

$\square$

Next we show that the Gegenbauer kernels can project any function into the space of their corresponding spherical harmonics,

**Lemma 3** (A Reproducing Kernel for $\mathcal{H}_\ell\left(\mathbb{S}^{d-1}\right)$). *For every $f \in L^2\left(\mathbb{S}^{d-1}\right)$, if $f = \sum_{\ell=0}^\infty f_\ell$ is the unique decomposition of $f$ over $\bigoplus_{\ell=0}^\infty \mathcal{H}_\ell\left(\mathbb{S}^{d-1}\right)$ as per Lemma 2, then $f_\ell$ is given by*

$$f_\ell(\sigma) = \alpha_{\ell,d} \cdot \mathop{\mathbb{E}}_{w\sim\mathcal{U}(\mathbb{S}^{d-1})}\left[f(w)P_d^\ell\left(\langle\sigma, w\rangle\right)\right] \quad \text{for } \sigma \in \mathbb{S}^{d-1}.$$

*Proof.* This is a classic textbook result, see [Mor98].

$\square$

Now we prove that the Gegenbauer kernels satisfy the reproducing property for the Hilbert space $\mathcal{H}_\ell(\mathbb{S}^{d-1})$.

**Lemma 1** (Reproducing Property of Zonal Harmonics). *Let $P_d^\ell(\cdot)$ be the Gengenbauer polynomial of degree $\ell$ in dimension $d$. For any $x, y \in \mathbb{S}^{d-1}$:*

$$P_d^\ell(\langle x, y\rangle) = \alpha_{\ell,d} \cdot \mathbb{E}_{w\sim\mathcal{U}(\mathbb{S}^{d-1})}\left[P_d^\ell\left(\langle x, w\rangle\right) P_d^\ell\left(\langle y, w\rangle\right)\right],$$

*Furthermore, for any $\ell' \neq \ell$:*

$$\mathbb{E}_{w\sim\mathcal{U}(\mathbb{S}^{d-1})}\left[P_d^\ell\left(\langle x, w\rangle\right) \cdot P_d^{\ell'}\left(\langle y, w\rangle\right)\right] = 0.$$

*Proof.* This result follows directly from the Funk–Hecke formula (See [AH12]). However, we provide another proof here. First note that for every $x \in \mathbb{S}^{d-1}$ the function $P_d^\ell \left( \langle x, \cdot \rangle \right) \in \mathcal{H}_\ell \left( \mathbb{S}^{d-1} \right)$. Therefore the first claim follow by applying Lemma 3 on function $f(\sigma) = P_d^\ell \left( \langle x, \sigma \rangle \right)$ which also satisfies $f_\ell = f$. On the other hand, $P_d^{\ell'} \left( \langle y, \cdot \rangle \right) \in \mathcal{H}_{\ell'} \left( \mathbb{S}^{d-1} \right)$ for every $y \in \mathbb{S}^{d-1}$. Thus, for $\ell' \neq \ell$, using the fact that spherical harmonics are orthogonal spaces of functions, $P_d^{\ell'} \left( \langle y, \cdot \rangle \right) \perp \mathcal{H}_\ell \left( \mathbb{S}^{d-1} \right)$, which gives the second claim.

$\square$

Next we prove that the kernel operator defined in Definition 2 is in fact a projection operator,

**Claim 1.** *The operator $\mathcal{K}_d^{(q)}$ defined in Definition 2 satisfies the property $\left( \mathcal{K}_d^{(q)} \right)^2 = \mathcal{K}_d^{(q)}$.*

*Proof.* For every $f \in L^2 \left( \mathbb{S}^{d-1} \right)$ and every $\sigma \in \mathbb{S}^{d-1}$, using Definition 2 we have,

$$
\left[ \left( \mathcal{K}_d^{(q)} \right)^2 f \right] (\sigma) = \sum_{\ell'=0}^q \frac{\alpha_{\ell',d}}{|\mathbb{S}^{d-1}|} \left\langle \mathcal{K}_d^{(q)} f, P_d^{\ell'} \left( \langle \sigma, \cdot \rangle \right) \right\rangle_{\mathbb{S}^{d-1}}
$$

$$
= \sum_{\ell=0}^q \sum_{\ell'=0}^q \alpha_{\ell,d} \alpha_{\ell',d} \cdot \mathbb{E}_{w \sim \mathcal{U}(\mathbb{S}^{d-1})} \left[ P_d^{\ell'} \left( \langle \sigma, w \rangle \right) \cdot \mathbb{E}_{\tau \sim \mathcal{U}(\mathbb{S}^{d-1})} \left[ P_d^\ell \left( \langle \tau, w \rangle \right) f(\tau) \right] \right]
$$

$$
= \sum_{\ell=0}^q \sum_{\ell'=0}^q \alpha_{\ell,d} \alpha_{\ell',d} \cdot \mathbb{E}_{\tau \sim \mathcal{U}(\mathbb{S}^{d-1})} \left[ f(\tau) \cdot \mathbb{E}_{w \sim \mathcal{U}(\mathbb{S}^{d-1})} \left[ P_d^{\ell'} \left( \langle \sigma, w \rangle \right) P_d^\ell \left( \langle \tau, w \rangle \right) \right] \right]
$$

$$
= \sum_{\ell=0}^q \alpha_{\ell,d} \cdot \mathbb{E}_{\tau \sim \mathcal{U}(\mathbb{S}^{d-1})} \left[ f(\tau) \cdot P_d^\ell \left( \langle \sigma, \tau \rangle \right) \right]
$$

$$
= \left[ \mathcal{K}_d^{(q)} f \right] (\sigma),
$$

where the fourth line above follows from Lemma 1. This proves the claim.

$\square$

# B  Reducing the Interpolation Problem to a Least-Squares Regression

In this section we show that our spherical harmonic interpolation problem, i.e., Problem 2, can be solved by approximately solving a least-squares problem as claimed in Claim 2. We start by showing that for any function $f \in L^2(\mathbb{S}^{d-1})$, $\mathcal{K}_d^{(q)} f$ gives its low-degree component. More precisely, let $f = \sum_{\ell=0}^\infty f_\ell$ be the decomposition of $f$ over the Hilbert sum $\bigoplus_{\ell=0}^\infty \mathcal{H}_\ell \left( \mathbb{S}^{d-1} \right)$ as per Lemma 2. Now if we let $\mathcal{K}_d^{(q)}$ be the kernel operator from Definition 2 and if $\left\{ y_1^\ell, y_2^\ell, \ldots, y_{\alpha_{\ell,d}}^\ell \right\}$ is an orthonormal basis for $\mathcal{H}_\ell \left( \mathbb{S}^{d-1} \right)$, then by Theorem 3 we have,

$$
\left[ \mathcal{K}_d^{(q)} f \right] (\sigma) = \sum_{\ell=0}^q \alpha_{\ell,d} \cdot \mathbb{E}_{w \sim \mathcal{U}(\mathbb{S}^{d-1})} \left[ f(w) P_d^\ell \left( \langle \sigma, w \rangle \right) \right]
$$

$$
= \sum_{\ell=0}^q |\mathbb{S}^{d-1}| \cdot \mathbb{E}_{w \sim \mathcal{U}(\mathbb{S}^{d-1})} \left[ f(w) \cdot \sum_{j=1}^{\alpha_{\ell,d}} y_j^\ell(\sigma) \cdot y_j^\ell(w) \right]
$$

$$
= \sum_{\ell=0}^q \sum_{j=1}^{\alpha_{\ell,d}} y_j^\ell(\sigma) \cdot |\mathbb{S}^{d-1}| \cdot \mathbb{E}_{w \sim \mathcal{U}(\mathbb{S}^{d-1})} \left[ f(w) \cdot y_j^\ell(w) \right]
$$

$$
= \sum_{\ell=0}^q \sum_{j=1}^{\alpha_{\ell,d}} \langle f, y_j^\ell(w) \rangle_{\mathbb{S}^{d-1}} \cdot y_j^\ell(\sigma)
$$

$$
= \sum_{\ell=0}^q f_\ell(\sigma) = f^{(q)}(\sigma),
$$

where the the second line above follows from Theorem 3, the fourth line follows from Eq. (2), and the last line follows from Lemma 2. This proves that the low-degree component $f^{(q)} = \mathcal{K}_d^{(q)} f$.

**Claim 2.** *For any $f \in L^2\left(\mathbb{S}^{d-1}\right)$, any integer $q \geq 0$, and any $C \geq 1$, if $\tilde{g} \in L^2\left(\mathbb{S}^{d-1}\right)$ satisfies,*

$$\left\|\mathcal{K}_d^{(q)} \tilde{g} - f\right\|_{\mathbb{S}^{d-1}}^2 \leq C \cdot \min_{g \in L^2(\mathbb{S}^{d-1})} \left\|\mathcal{K}_d^{(q)} g - f\right\|_{\mathbb{S}^{d-1}}^2 ,$$

*and if we let $f^{(q)} := \mathcal{K}_d^{(q)} f$, where $\mathcal{K}_d^{(q)}$ is defined as per Definition 2, then the following holds*

$$\left\|\mathcal{K}_d^{(q)} \tilde{g} - f^{(q)}\right\|_{\mathbb{S}^{d-1}}^2 \leq (C - 1) \cdot \left\|f^{(q)} - f\right\|_{\mathbb{S}^{d-1}}^2 .$$

*Proof.* First, note that $g^* = f$ is an optimal solution to the least-squares problem in Eq. (7). Thus we have,

$$\min_{g \in L^2(\mathbb{S}^{d-1})} \left\|\mathcal{K}_d^{(q)} g - f\right\|_{\mathbb{S}^{d-1}}^2 = \left\|\mathcal{K}_d^{(q)} f - f\right\|_{\mathbb{S}^{d-1}}^2 = \left\|f^{(q)} - f\right\|_{\mathbb{S}^{d-1}}^2 .$$

Next, we can write,

$$\begin{aligned}
\left\|\mathcal{K}_d^{(q)} \tilde{g} - f\right\|_{\mathbb{S}^{d-1}}^2 &= \left\|\mathcal{K}_d^{(q)} \tilde{g} - \mathcal{K}_d^{(q)} f + \left(\mathcal{K}_d^{(q)} f - f\right)\right\|_{\mathbb{S}^{d-1}}^2 \\
&= \left\|\mathcal{K}_d^{(q)} (\tilde{g} - f) + \left(\mathcal{K}_d^{(q)} f - f\right)\right\|_{\mathbb{S}^{d-1}}^2 \\
&= \left\|\mathcal{K}_d^{(q)} (\tilde{g} - f)\right\|_{\mathbb{S}^{d-1}}^2 + \left\|\mathcal{K}_d^{(q)} f - f\right\|_{\mathbb{S}^{d-1}}^2 \\
&= \left\|\mathcal{K}_d^{(q)} \tilde{g} - f^{(q)}\right\|_{\mathbb{S}^{d-1}}^2 + \left\|f^{(q)} - f\right\|_{\mathbb{S}^{d-1}}^2 ,
\end{aligned}$$

where the third line follows from the Pythagorean theorem because $\mathcal{K}_d^{(q)} (\tilde{g} - f) \in \mathcal{H}^{(q)}\left(\mathbb{S}^{d-1}\right)$ while $\mathcal{K}_d^{(q)} f - f = -\sum_{\ell > q} f_\ell$, thus $\left(\mathcal{K}_d^{(q)} f - f\right) \perp \mathcal{H}^{(q)}\left(\mathbb{S}^{d-1}\right)$. Combining the two equalities above with inequality $\left\|\mathcal{K}_d^{(q)} \tilde{g} - f\right\|_{\mathbb{S}^{d-1}}^2 \leq C \cdot \min_{g \in L^2(\mathbb{S}^{d-1})} \left\|\mathcal{K}_d^{(q)} g - f\right\|_{\mathbb{S}^{d-1}}^2$ given in the statement of the claim, proves Claim 2.

$\square$

## C  Approximate Regression via Leverage Score Sampling

In this section we ultimately prove our main result of Theorem 4. We start by proving useful properties of the leverage function given in Definition 3. First, we show the fact that the leverage function can be characterized in terms of a least-squares minimization problem, which is crucial for computing the leverage scores distribution. This fact was previously exploited in [AKM+17] and [AKM+19] in the context of Fourier operators.

**Lemma 4** (Min Characterization of the Leverage Function). *For any $w \in \mathbb{S}^{d-1}$, let $\tau_q(w)$ be the leverage function (Definition 3) and define $\phi_w \in L^2(\mathbb{S}^{d-1})$ by $\phi_w(\sigma) \equiv \sum_{\ell=0}^{q} \frac{\alpha_{\ell,d}}{|\mathbb{S}^{d-1}|} P_d^\ell (\langle \sigma, w \rangle)$. We have the following minimization characterization of the leverage function:*

$$\tau_q(w) = \left\{ \min_{g \in L^2(\mathbb{S}^{d-1})} \|g\|_{\mathbb{S}^{d-1}}^2, \quad s.t. \ \mathcal{K}_d^{(q)} g = \phi_w \right\}. \tag{9}$$

We remark that this lemma is in fact an adaptation and generalization of Theorem 5 of [AKM+19]. We prove this lemma here for the sake of completeness.

*Proof.* First we show that the right hand side of Eq. (9) is never smaller than the leverage function in Definition 3. Let $g_w^* \in L^2(\mathbb{S}^{d-1})$ be the optimal solution of Eq. (9) for any $w \in \mathbb{S}^{d-1}$. Note that the

optimal solution satisfies $\mathcal{K}_d^{(q)} g_w^* = \phi_w$. Thus, for any function $f \in L^2(\mathbb{S}^{d-1})$, using Definition 2, we can write

$$
\left| \left[ \mathcal{K}_d^{(q)} f \right](w) \right|^2 = \left| \sum_{\ell=0}^{q} \alpha_{\ell,d} \cdot \mathop{\mathbb{E}}_{\sigma \sim \mathcal{U}(\mathbb{S}^{d-1})} \left[ P_d^\ell \left( \langle \sigma, w \rangle \right) \cdot f(\sigma) \right] \right|^2
$$

$$
= |\langle \phi_w, f \rangle_{\mathbb{S}^{d-1}}|^2 = \left| \left\langle \mathcal{K}_d^{(q)} g_w^*, f \right\rangle_{\mathbb{S}^{d-1}} \right|^2
$$

$$
= \left| \left\langle g_w^*, \mathcal{K}_d^{(q)} f \right\rangle_{\mathbb{S}^{d-1}} \right|^2 \qquad \text{(because } \mathcal{K}_d^{(q)} \text{ is self-adjoint)}
$$

$$
\leq \|g_w^*\|_{\mathbb{S}^{d-1}}^2 \cdot \left\| \mathcal{K}_d^{(q)} f \right\|_{\mathbb{S}^{d-1}}^2 \qquad \text{(by Cauchy–Schwarz inequality)}
$$

Therefore, for any $f \in L^2(\mathbb{S}^{d-1})$ with $\left\| \mathcal{K}_d^{(q)} f \right\|_{\mathbb{S}^{d-1}} > 0$, we have

$$
\frac{\left| \left[ \mathcal{K}_d^{(q)} f \right](w) \right|^2}{\left\| \mathcal{K}_d^{(q)} f \right\|_{\mathbb{S}^{d-1}}^2} \leq \|g_w^*\|_{\mathbb{S}^{d-1}}^2. \tag{11}
$$

We conclude the proof by showing that the maximum value is attained. First, we show that the optimal solution $g_w^*$ of Eq. (9) satisfies the property that $\mathcal{K}_d^{(q)} g_w^* = g_w^*$. Suppose for the sake of contradiction that $\mathcal{K}_d^{(q)} g_w^* \neq g_w^*$. In this case, Claim 1 implies that,

$$
\mathcal{K}_d^{(q)} \left( \mathcal{K}_d^{(q)} g_w^* - g_w^* \right) = \left( \mathcal{K}_d^{(q)} \right)^2 g_w^* - \mathcal{K}_d^{(q)} g_w^* = \mathcal{K}_d^{(q)} g_w^* - \mathcal{K}_d^{(q)} g_w^* = 0.
$$

Thus, the function $g = \mathcal{K}_d^{(q)} g_w^*$ satisfies the constraint of the minimization problem in Eq. (9). Now, using the above and the fact that $\mathcal{K}_d^{(q)}$ is self-adjoint we can write,

$$
\left\langle \mathcal{K}_d^{(q)} g_w^*, \mathcal{K}_d^{(q)} g_w^* - g_w^* \right\rangle_{\mathbb{S}^{d-1}} = \left\langle g_w^*, \mathcal{K}_d^{(q)} \left( \mathcal{K}_d^{(q)} g_w^* - g_w^* \right) \right\rangle_{\mathbb{S}^{d-1}} = 0.
$$

This shows that $\mathcal{K}_d^{(q)} g_w^* \perp \left( \mathcal{K}_d^{(q)} g_w^* - g_w^* \right)$, hence by Pythagorean theorem we have,

$$
\|g_w^*\|_{\mathbb{S}^{d-1}}^2 = \left\| \mathcal{K}_d^{(q)} g_w^* \right\|_{\mathbb{S}^{d-1}}^2 + \left\| \mathcal{K}_d^{(q)} g_w^* - g_w^* \right\|_{\mathbb{S}^{d-1}}^2 > \left\| \mathcal{K}_d^{(q)} g_w^* \right\|_{\mathbb{S}^{d-1}}^2 = \|g\|_{\mathbb{S}^{d-1}}^2 ,
$$

which is in contrast with the assumption that $g_w^*$ is the optimal solution of Eq. (9). Therefore, our claim that $\mathcal{K}_d^{(\ell)} g_w^* = g_w^*$ holds.

Now, we show that for $f = g_w^*$, the maximum value in inequality Eq. (11) is attained. For any $w \in \mathbb{S}^{d-1}$ we have the following

$$
\left[ \mathcal{K}_d^{(q)} f \right](w) = \left\langle \mathcal{K}_d^{(q)} g_w^*, f \right\rangle_{\mathbb{S}^{d-1}} = \langle g_w^*, g_w^* \rangle_{\mathbb{S}^{d-1}} = \|g_w^*\|_{\mathbb{S}^{d-1}}^2.
$$

On the other hand we have $\left\| \mathcal{K}_d^{(q)} f \right\|_{\mathbb{S}^{d-1}}^2 = \|g_w^*\|_{\mathbb{S}^{d-1}}^2$. Thus, $\left\| \mathcal{K}_d^{(q)} f \right\|_{\mathbb{S}^{d-1}}^{-2} \cdot \left| \left[ \mathcal{K}_d^{(q)} f \right](w) \right|^2 = \|g_w^*\|_{\mathbb{S}^{d-1}}^2$ which implies that $\tau_q(w) = \|g_w^*\|_{\mathbb{S}^{d-1}}^2$ and thus proves the lemma.

$\square$

Next we prove that the leverage function is constant.

**Lemma 5** (Leverage Function is Constant). *The leverage function given in Definition 3 is equal to* $\tau_q(w) = \sum_{\ell=0}^{q} \frac{\alpha_{\ell,d}}{|\mathbb{S}^{d-1}|}$ *for every* $w \in \mathbb{S}^{d-1}$.

*Proof.* First we prove that $\tau_q(w) \leq \sum_{\ell=0}^{q} \frac{\alpha_{\ell,d}}{|\mathbb{S}^{d-1}|}$ using the min-characterization. If we let $\phi_w \in L^2(\mathbb{S}^{d-1})$ be defined as $\phi_w(\sigma) := \sum_{\ell=0}^{q} \frac{\alpha_{\ell,d}}{|\mathbb{S}^{d-1}|} P_d^\ell \left( \langle \sigma, w \rangle \right)$, then by Definition 2, for every $\sigma \in \mathbb{S}^{d-1}$

we can write,

$$\left[\mathcal{K}_d^{(q)}\phi_w\right](\sigma) = \sum_{\ell=0}^{q}\alpha_{\ell,d}\cdot\mathop{\mathbb{E}}_{v\sim\mathcal{U}(\mathbb{S}^{d-1})}\left[P_d^\ell\left(\langle\sigma,v\rangle\right)\cdot\phi_w(v)\right]$$

$$= \sum_{\ell=0}^{q}\sum_{\ell'=0}^{q}\frac{\alpha_{\ell,d}\alpha_{\ell',d}}{|\mathbb{S}^{d-1}|}\cdot\mathop{\mathbb{E}}_{v\sim\mathcal{U}(\mathbb{S}^{d-1})}\left[P_d^\ell\left(\langle\sigma,v\rangle\right)\cdot P_d^{\ell'}\left(\langle v,w\rangle\right)\right]$$

$$= \sum_{\ell=0}^{q}\frac{\alpha_{\ell,d}}{|\mathbb{S}^{d-1}|}P_d^\ell\left(\langle\sigma,w\rangle\right) = \phi_w(\sigma), \tag{12}$$

where the third line above follows from Lemma 1. Therefore, the test function $g := \phi_w$ satisfies the constraint of the minimization in Eq. (9), i.e., $\mathcal{K}_d^{(q)}g = \phi_w$. Thus, Lemma 4 implies that,

$$\tau_q(w) \leq \|g\|_{\mathbb{S}^{d-1}}^2 = \|\phi_w\|_{\mathbb{S}^{d-1}}^2 = \sum_{\ell=0}^{q}\frac{\alpha_{\ell,d}}{|\mathbb{S}^{d-1}|},$$

where the equality above follows from Lemma 1 along with Eq. (2). This establishes the upper bound on the leverage function that we sought to prove.

Now, using the maximization characterization of the leverage function in Definition 3, we prove that $\tau_q(w) \geq \sum_{\ell=0}^{q}\frac{\alpha_{\ell,d}}{|\mathbb{S}^{d-1}|}$. Again, we consider the same test function $g = \phi_w$ and write,

$$\left\|\mathcal{K}_d^{(q)}\phi_w\right\|_{\mathbb{S}^{d-1}}^{-2}\cdot\left|\left[\mathcal{K}_d^{(q)}\phi_w\right](w)\right|^2 = \frac{|\phi_w(w)|^2}{\|\phi_w\|_{\mathbb{S}^{d-1}}^2}$$

$$= \frac{\left|\sum_{\ell=0}^{q}\frac{\alpha_{\ell,d}}{|\mathbb{S}^{d-1}|}P_d^\ell\left(\langle w,w\rangle\right)\right|^2}{\sum_{\ell=0}^{q}\frac{\alpha_{\ell,d}}{|\mathbb{S}^{d-1}|}}$$

$$= \frac{\left|\sum_{\ell=0}^{q}\frac{\alpha_{\ell,d}}{|\mathbb{S}^{d-1}|}P_d^\ell(1)\right|^2}{\sum_{\ell=0}^{q}\frac{\alpha_{\ell,d}}{|\mathbb{S}^{d-1}|}} = \sum_{\ell=0}^{q}\frac{\alpha_{\ell,d}}{|\mathbb{S}^{d-1}|},$$

where the first and second line above follow from Eq. (12) and Lemma 1, respectively. Therefore, the max characterization of the leverage function in Definition 3 implies that,

$$\tau_q(w) \geq \left\|\mathcal{K}_d^{(q)}\phi_w\right\|_{\mathbb{S}^{d-1}}^{-2}\cdot\left|\left[\mathcal{K}_d^{(q)}\phi_w\right](w)\right|^2 = \sum_{\ell=0}^{q}\frac{\alpha_{\ell,d}}{|\mathbb{S}^{d-1}|}.$$

This completes the proof of Lemma 5 and establishes that $\tau_q(w)$ is uniformly equal to $\sum_{\ell=0}^{q}\frac{\alpha_{\ell,d}}{|\mathbb{S}^{d-1}|}$.

$\square$

To prove Theorem 4, we need to use prior results about solving linear systems in continuous setting via leverage score sampling. In particular, we use Theorem 6.3 from [CP19], which is restated below,

**Theorem 7** (Theorem 6.3 of [CP19]). *Consider any dimension $n$ linear space $\mathcal{F}$ of functions from a domain $G$ to $\mathbb{R}$. Let $D$ be a distribution over $G$ and $f$ be some function from $G$ to $\mathbb{R}$. Also, define the norm with respect to $D$ of any function $h : G \to \mathbb{R}$ as $\|h\|_D^2 := \mathbb{E}_{x\sim D}[|h(x)|^2]$ and let $y = \arg\min_{h\in\mathcal{F}}\|h - f\|_D^2$. Fix any distribution $D'$ over $G$ and let $K_{D'} := \sup_{x\in G}\sup_{h\in\mathcal{F}}\left\{\frac{D(x)}{D'(x)}\cdot\frac{|h(x)|^2}{\|h\|_D^2}\right\}$.*

*For i.i.d. sample query points $x_1, x_2, \ldots x_s \sim D'$ and weights $w_i = \frac{D(x_i)}{s\cdot D'(x_i)}$ for $i \in [s]$, let the weighted ERM estimator $\widetilde{f}_s$ be defined as $\widetilde{f}_s := \arg\min_{h\in\mathcal{F}}\sum_{i=1}^{s}w_i\cdot|h(x_i) - f(x_i)|^2$. For any $\varepsilon > 0$ and a sufficiently large fixed constant $c$, if the number of queries $s \geq c\cdot\left(K_{D'}\log n + \frac{K_{D'}}{\varepsilon}\right)$, then the weighted ERM estimator $\widetilde{f}_s$ satisfies,*

$$\Pr\left[\left\|\widetilde{f}_s - y\right\|_D^2 \leq \varepsilon\cdot\|f - y\|_D^2\right] \geq 1 - 10^{-4}.$$

Now we are ready to prove Theorem 4. Our approach is to apply Theorem 7 to the space of degree $q$ spherical harmonics $\mathcal{H}^{(q)}\left(\mathbb{S}^{d-1}\right)$ and use the fact that the leverage scores distribution of the operator $\mathcal{K}_d^{(q)}$ are uniform on the unit sphere $\mathbb{S}^{d-1}$.

**Theorem 4** (Approximate Regression via Leverage Function Sampling). *For any $\varepsilon > 0$, let $s = c \cdot \left( \beta_{q,d} \log \beta_{q,d} + \frac{\beta_{q,d}}{\varepsilon} \right)$, for sufficiently large fixed constant c, and let $x_1, x_2, \ldots, x_s$ be i.i.d. uniform samples on $\mathbb{S}^{d-1}$. Define the quasi-matrix $\boldsymbol{P} : \mathbb{R}^s \to L^2(\mathbb{S}^{d-1})$ as follows, for every $v \in \mathbb{R}^d$:*

$$[\boldsymbol{P}\, v](\sigma) := \sum_{\ell=0}^{q} \frac{\alpha_{\ell,d}}{\sqrt{s} \cdot |\mathbb{S}^{d-1}|} \cdot \sum_{j=1}^{s} v_j \cdot P_d^{\ell}\left(\langle x_j, \sigma \rangle\right) \quad for\ \sigma \in \mathbb{S}^{d-1}.$$

*Also let $\boldsymbol{f} \in \mathbb{R}^s$ be a vector with $\boldsymbol{f}_j := \frac{1}{\sqrt{s}} \cdot f(x_j)$ for $j = 1, 2, \ldots, s$ and let $\boldsymbol{P}^*$ be the adjoint of $\boldsymbol{P}$. If $\tilde{g}$ is an optimal solution to the least-squares problem $\tilde{g} \in \arg\min_{g \in L^2(\mathbb{S}^{d-1})} \|\boldsymbol{P}^*g - \boldsymbol{f}\|_2^2$, then with probability at least $1 - 10^{-4}$ the following holds,*

$$\left\| \mathcal{K}_d^{(q)} \tilde{g} - f \right\|_{\mathbb{S}^{d-1}}^2 \le (1 + \varepsilon) \cdot \min_{g \in L^2(\mathbb{S}^{d-1})} \left\| \mathcal{K}_d^{(q)} g - f \right\|_{\mathbb{S}^{d-1}}^2.$$

*Proof.* We prove this theorem by invoking Theorem 7. To do so, we first let the space $\mathcal{F}$ of function from $\mathbb{S}^{d-1}$ to $\mathbb{R}$ be $\mathcal{F} := \mathcal{H}^{(q)}\left(\mathbb{S}^{d-1}\right)$. It is clear that the space of spherical harmonics is a linear space of functions because of the existence of the kernel operator $\mathcal{K}_d^{(q)}$ which is a projection operator onto $\mathcal{H}^{(q)}\left(\mathbb{S}^{d-1}\right)$, so $\mathcal{F}$ satisfies the first precondition of Theorem 7. Additionally, the dimension of this space of functions is $n = \beta_{q,d}$.

Also, let $D$ be a uniform distribution over the unit sphere $\mathbb{S}^{d-1}$. For this distribution, the norm defined in Theorem 7 satisfies $\|h\|_{\mathbb{S}^{d-1}}^2 = |\mathbb{S}^{d-1}| \cdot \|h\|_D^2$, for any $h \in L^2(\mathbb{S}^{d-1})$. Moreover, let $D'$ be a uniform distribution over the unit sphere $\mathbb{S}^{d-1}$ as well.

Now we show that for these choices of $\mathcal{F}, D, D'$, the condition number $K_{D'}$ defined as per Theorem 7 is equal to $\beta_{q,d}$. We can write,

$$K_{D'} := \sup_{x \in \mathbb{S}^{d-1}} \sup_{h \in \mathcal{F}} \left\{ \frac{D(x)}{D'(x)} \cdot \frac{|h(x)|^2}{\|h\|_D^2} \right\}$$

$$= |\mathbb{S}^{d-1}| \cdot \sup_{x \in \mathbb{S}^{d-1}} \sup_{h \in \mathcal{F}} \left\{ \frac{|h(x)|^2}{\|h\|_{\mathbb{S}^{d-1}}^2} \right\}$$

$$= |\mathbb{S}^{d-1}| \cdot \sup_{x \in \mathbb{S}^{d-1}} \sup_{g \in L^2(\mathbb{S}^{d-1})} \left\{ \frac{\left| \left[ \mathcal{K}_d^{(q)} g \right](x) \right|^2}{\left\| \mathcal{K}_d^{(q)} g \right\|_{\mathbb{S}^{d-1}}^2} \right\}$$

$$= |\mathbb{S}^{d-1}| \cdot \sup_{x \in \mathbb{S}^{d-1}} \tau_q(x)$$

$$= \beta_{q,d},$$

where the second line above follows from the fact that $D, D'$ are both equal to the uniform distribution over $\mathbb{S}^{d-1}$ and $\|h\|_{\mathbb{S}^{d-1}}^2 = |\mathbb{S}^{d-1}| \cdot \|h\|_D^2$. The third line above follows from the fact that any function $h \in \mathcal{H}^{(q)}\left(\mathbb{S}^{d-1}\right)$ can be expressed as $h = \mathcal{K}_d^{(q)} g$ for some $g \in L^2(\mathbb{S}^{d-1})$ because $\mathcal{K}_d^{(q)}$ is the projection operator onto $\mathcal{H}^{(q)}\left(\mathbb{S}^{d-1}\right)$. The fourth line follows from Definition 3 and last line follows from Lemma 5.

Finally, in order to invoke Theorem 7, we can write that the weighted ERM estimator $\widetilde{f}_s$ is equal to the following,

$$
\begin{aligned}
\widetilde{f}_s &:= \arg\min_{h \in \mathcal{F}} \sum_{i=1}^{s} w_i \cdot |h(x_i) - f(x_i)|^2 \\
&= \arg\min_{g \in L^2(\mathbb{S}^{d-1})} \sum_{i=1}^{s} w_i \cdot \left| \left[ \mathcal{K}_d^{(q)} g \right](x_i) - f(x_i) \right|^2 \\
&= \arg\min_{g \in L^2(\mathbb{S}^{d-1})} \sum_{i=1}^{s} \left| \frac{1}{\sqrt{s}} \cdot \left[ \mathcal{K}_d^{(q)} g \right](x_i) - \boldsymbol{f}_i \right|^2 \\
&= \arg\min_{g \in L^2(\mathbb{S}^{d-1})} \| \boldsymbol{P}^* g - \boldsymbol{f} \|_2^2,
\end{aligned}
$$

where the third line above uses the definition of $\boldsymbol{f}_i = \frac{1}{\sqrt{s}} f(x_i)$, and the last line follows from the definition of adjoint of the quasi-matrix $\boldsymbol{P}$. Therefore, the theorem follows by invoking Theorem 7.

$\square$

## D  Efficient Algorithm for Spherical Harmonic Interpolation

In this section we prove our main theorem about our spherical harmonic interpolation algorithm.

**Theorem 5** (Efficient Spherical Harmonic Interpolation). *Algorithm 1 returns a function $y \in \mathcal{H}^{(q)}(\mathbb{S}^{d-1})$ such that, with probability at least $1 - 10^{-4}$:*

$$
\left\| y - f^{(q)} \right\|_{\mathbb{S}^{d-1}}^2 \le \varepsilon \cdot \left\| f^{(q)} - f \right\|_{\mathbb{S}^{d-1}}^2, \quad \text{where } f^{(q)} := \mathcal{K}_d^{(q)} f.
$$

*Suppose we can compute the Gegenbauer polynomial $P_d^\ell(t)$ at every point $t \in [-1, 1]$ in constant time. Then Algorithm 1 queries the function $f$ at $s = \mathcal{O}\left( \beta_{q,d} \log \beta_{q,d} + \frac{\beta_{q,d}}{\varepsilon} \right)$ points on the sphere $\mathbb{S}^{d-1}$ and runs in $\mathcal{O}(s^2 \cdot d + s^\omega)$ time. This algorithm evaluates $y(\sigma)$ in $\mathcal{O}(d \cdot s)$ time for any $\sigma \in \mathbb{S}^{d-1}$.*

*Proof.* First note that the random points $w_1, w_2, \ldots, w_s$ in line 3 of Algorithm 1 are i.i.d. sample with uniform distribution on the surface of $\mathbb{S}^{d-1}$. Therefore, we can invoke Theorem 4. More specifically, if we let $\boldsymbol{P}$ be the quasi-matrix defined in Theorem 4 corresponding to the random points $w_1, w_2, \ldots, w_s$ sampled in line 3 and if we let $\boldsymbol{f}$ be the vector of function samples defined in line 5 of the algorithm, then with probability at least $1 - 10^{-4}$, any optimal solution to the following least-squares problem

$$
\tilde{g} \in \arg\min_{g \in L^2(\mathbb{S}^{d-1})} \| \boldsymbol{P}^* g - \boldsymbol{f} \|_2^2, \tag{13}
$$

satisfies the following,

$$
\left\| \mathcal{K}_d^{(q)} \tilde{g} - f \right\|_{\mathbb{S}^{d-1}}^2 \le (1 + \varepsilon) \cdot \min_{g \in L^2(\mathbb{S}^{d-1})} \left\| \mathcal{K}_d^{(q)} g - f \right\|_{\mathbb{S}^{d-1}}^2. \tag{14}
$$

Now note that the least-squares problem in Eq. (13) has at least one optimal solution $\tilde{g}$ which is in the eigenspace of the operator $\boldsymbol{P}\boldsymbol{P}^*$. More specifically, there exists a vector $\boldsymbol{z} \in \mathbb{R}^s$ such that $\tilde{g} = \boldsymbol{P} \cdot \boldsymbol{z}$ is an optimal solution for Eq. (13). Therefore, we can focus on finding this optimal solution by solving the following least-squares problem

$$
\boldsymbol{z} \in \arg\min_{\boldsymbol{x} \in \mathbb{R}^s} \| \boldsymbol{P}^* \boldsymbol{P} \boldsymbol{x} - \boldsymbol{f} \|_2^2,
$$

and then letting $\tilde{g} = \boldsymbol{P} \cdot \boldsymbol{z}$. This $\tilde{g}$ is guaranteed to be an optimal solution for Eq. (13), thus it satisfies Eq. (14). We solve the above least-squares problem using the kernel trick. In fact we show that $\boldsymbol{P}^* \boldsymbol{P}$ is equal to the kernel matrix $\boldsymbol{K}$ computed in line 4 of Algorithm 1. To see why, note that for any

$i, j \in [s]$ we have,

$$
\begin{aligned}
\left[\boldsymbol{P}^*\boldsymbol{P}\right]_{i,j} &= \left\langle \sum_{\ell=0}^{q} \frac{\alpha_{\ell,d}}{\sqrt{s \cdot |\mathbb{S}^{d-1}|}} \cdot P_d^\ell\left(\langle w_i, \cdot \rangle\right), \sum_{\ell=0}^{q} \frac{\alpha_{\ell,d}}{\sqrt{s \cdot |\mathbb{S}^{d-1}|}} \cdot P_d^\ell\left(\langle w_j, \cdot \rangle\right) \right\rangle_{\mathbb{S}^{d-1}} \\
&= \sum_{\ell=0}^{q} \sum_{\ell'=0}^{q} \frac{\alpha_{\ell,d}\alpha_{\ell',d}}{s \cdot |\mathbb{S}^{d-1}|^2} \cdot \left\langle P_d^\ell\left(\langle w_i, \cdot \rangle\right), P_d^{\ell'}\left(\langle w_j, \cdot \rangle\right) \right\rangle_{\mathbb{S}^{d-1}} \\
&= \sum_{\ell=0}^{q} \sum_{\ell'=0}^{q} \frac{\alpha_{\ell,d}\alpha_{\ell',d}}{s \cdot |\mathbb{S}^{d-1}|} \cdot \mathop{\mathbb{E}}_{v \sim \mathcal{U}(\mathbb{S}^{d-1})} \left[ P_d^\ell\left(\langle w_i, v \rangle\right) \cdot P_d^{\ell'}\left(\langle w_j, v \rangle\right) \right] \\
&= \sum_{\ell=0}^{q} \frac{\alpha_{\ell,d}}{s \cdot |\mathbb{S}^{d-1}|} \cdot P_d^\ell\left(\langle w_i, w_j \rangle\right) = \boldsymbol{K}_{i,j},
\end{aligned}
$$

where the fourth line above follows from Lemma 1. Therefore, we are interested in the optimal solution of the following least-squares problem

$$
\boldsymbol{z} \in \arg\min_{\boldsymbol{x} \in \mathbb{R}^s} \|\boldsymbol{K}\boldsymbol{x} - \boldsymbol{f}\|_2^2.
$$

The least-squares solution to the above problem is $\boldsymbol{z} = \boldsymbol{K}^\dagger \boldsymbol{f}$ which is exactly what is computed in line 6 of the algorithm. Now note that, the function $\tilde{g} = \boldsymbol{P} \cdot \boldsymbol{z}$ satisfies Eq. (14). Because $\tilde{g} = \boldsymbol{P} \cdot \boldsymbol{z} \in \mathcal{H}^{(q)}(\mathbb{S}^{d-1})$ and because $\mathcal{K}_d^{(q)}$ is an orthonormal projection operator into $\mathcal{H}^{(q)}(\mathbb{S}^{d-1})$, we have $\mathcal{K}_d^{(q)} \cdot \tilde{g} = \tilde{g} = \boldsymbol{P} \cdot \boldsymbol{z}$. This together with Eq. (14) imply that,

$$
\|\boldsymbol{P} \cdot \boldsymbol{z} - f\|_{\mathbb{S}^{d-1}}^2 \leq (1 + \varepsilon) \cdot \min_{g \in L^2(\mathbb{S}^{d-1})} \left\| \mathcal{K}_d^{(q)} g - f \right\|_{\mathbb{S}^{d-1}}^2.
$$

Now if we invoke Claim 2 with $C = 1 + \varepsilon$ on the above inequality we find that,

$$
\left\| \boldsymbol{P} \cdot \boldsymbol{z} - f^{(q)} \right\|_{\mathbb{S}^{d-1}}^2 \leq \varepsilon \cdot \left\| f^{(q)} - f \right\|_{\mathbb{S}^{d-1}}^2.
$$

Finally, one can easily see that the function $y \in \mathcal{H}^{(q)}(\mathbb{S}^{d-1})$ that Algorithm 1 outputs in line 7 is exactly equal to $y = \boldsymbol{P} \cdot \boldsymbol{z}$. This completes the accuracy bound of the theorem.

**Runtime and Sample Complexity.** these bounds follow from observing that:

- $s \cdot d$ time is needed to generate $w_1, w_2, \ldots, w_s$ in line 3 of the algorithm. To do this, we first generate random Gaussian points in $\mathbb{R}^d$ and then project then onto $\mathbb{S}^{d-1}$ by normalizing them.

- $s^2 \cdot d$ operations are needed to form the kernel matrix $\boldsymbol{K}$ in line 4 of the algorithm.

- $s$ queries to function $f$ are needed to form the samples vector $\boldsymbol{f}$ in line 5 of the algorithm.

- $s^\omega$ time is needed to compute the least-squares solution $\boldsymbol{z} = \boldsymbol{K}^\dagger \boldsymbol{f}$ in line 6 of the algorithm.

- $s \cdot d$ operations are needed to evaluate the output function $y(\sigma)$ in line 7 of the algorithm.

This completes the proof of Theorem 5. $\qquad\square$

# E  Lower Bound: Claims and Lemmas

In this section we prove the Claims and Lemmas used in our lower bound analysis for proving Theorem 6.

**Claim 4.** *Given the random input $f = \sum_{\ell=0}^{q} \boldsymbol{Y}_\ell \cdot v^{(\ell)}$ generated as described in Section 4, to solve Problem 2, an algorithm must return a function $\tilde{f}^{(q)} \in \mathcal{H}^{(q)}\left(\mathbb{S}^{d-1}\right)$ such that $\|\tilde{f}^{(q)} - f\|_{\mathbb{S}^{d-1}}^2 = 0$.*

*Proof.* Note that Problem 2 requires recovering a function $\tilde{f}^{(q)} \in \mathcal{H}^{(q)}\left(\mathbb{S}^{d-1}\right)$ such that:

$$\left\|\tilde{f}^{(q)} - f^{(q)}\right\|_{\mathbb{S}^{d-1}}^2 \leq \varepsilon \cdot \left\|f^{(q)} - f\right\|_{\mathbb{S}^{d-1}}^2, \tag{15}$$

where $f^{(q)} = \mathcal{K}_d^{(q)} f$. Using the definition of the input function $f = \sum_{\ell=0}^q \boldsymbol{Y}_\ell \cdot v^{(\ell)}$, we can write,

$$\begin{aligned}
f^{(q)} = \mathcal{K}_d^{(q)} f &= \sum_{\ell=0}^q \mathcal{K}_d^{(q)} \cdot \boldsymbol{Y}_\ell \cdot v^{(\ell)} \\
&= \sum_{\ell=0}^q \left(\sum_{\ell'=0}^q \boldsymbol{Y}_{\ell'} \boldsymbol{Y}_{\ell'}^*\right) \cdot \boldsymbol{Y}_\ell \cdot v^{(\ell)} \\
&= \sum_{\ell=0}^q \boldsymbol{Y}_\ell \cdot v^{(\ell)} = f,
\end{aligned}$$

where the equality in the second line above follows from Eq. (10) and the addition theorem in Theorem 3, and the third line follows because the operator $\boldsymbol{Y}_\ell$ has orthonormal columns and thus $\boldsymbol{Y}_{\ell'}^* \boldsymbol{Y}_\ell = I_{\alpha_{\ell,d}} \cdot \mathbb{1}_{\{\ell=\ell'\}}$. Therefore, plugging this into Eq. (15) gives,

$$\left\|\tilde{f}^{(q)} - f\right\|_{\mathbb{S}^{d-1}}^2 = \left\|\tilde{f}^{(q)} - f^{(q)}\right\|_{\mathbb{S}^{d-1}}^2 \leq \varepsilon \cdot \|f^{(q)} - f\|_{\mathbb{S}^{d-1}}^2 = \varepsilon \cdot \|f - f\|_{\mathbb{S}^{d-1}}^2 = 0.$$

$\square$

**Lemma 6.** *If a deterministic algorithm solves Problem 2 with probability at least $1/10$ over our random input distribution $f = \sum_{\ell=0}^q \boldsymbol{Y}_\ell \cdot v^{(\ell)}$, then with probability at least $1/10$, the output of the algorithm $\tilde{f}^{(q)}$ satisfies $\boldsymbol{Y}_\ell^* \tilde{f}^{(q)} = v^{(\ell)}$ for all integers $\ell \leq q$.*

*Proof.* By Claim 4, the output of the algorithm that solves Problem 2, satisfies $\left\|\tilde{f}^{(q)} - f\right\|_{\mathbb{S}^{d-1}}^2 = 0$. Therefore, by orthonormality of the columns of the operator $\boldsymbol{Y}_\ell$, we can write,

$$\boldsymbol{Y}_\ell^* \tilde{f}^{(q)} = \boldsymbol{Y}_\ell^* f + \boldsymbol{Y}_\ell^* (\tilde{f}^{(q)} - f) = \sum_{\ell'=0}^q \boldsymbol{Y}_\ell^* \boldsymbol{Y}_{\ell'} \cdot v^{(\ell')} = v^{(\ell)}.$$

$\square$

