# OpenReview forum: "Near Optimal Reconstruction of Spherical Harmonic Expansions"
_NeurIPS.cc/2023/Conference — NeurIPS 2023 poster_

### Official Review · Reviewer_L1Mq · 2023-07-01

**Soundness:** 3 good
**Presentation:** 2 fair
**Contribution:** 3 good
**Rating:** 6
**Confidence:** 3

**Summary:**

### Result ###

The paper studies the problem of recovering a function from a finite number of noisy observations, for the class of "square-integrable functions on the unit sphere" (denoted by $L^2(\mathbb{S}^{d-1})$ when the sphere in question is of degree $d$).

The paper shows, by developing an algorithm, that the number of samples required for recovery up to an $\epsilon$-multiplicative error is proportional to $\beta_{q, d}$, where $q$ is the degree of spherical harmonics desired and $\beta_{q, d}$ is the dimension of the space of degree $q$ spherical harmonics on the sphere of dimension $d-1$. This sample complexity is optimal, as shown by the paper via lower bounds.

---------------

### Broader Contributions ###

To achieve the result above, the paper makes the following conceptual and technical contributions.

First, the problem at hand is a regression problem with potentially infinite-dimensional continuous cost functions. Inspired by the approach of Avron, Kapralov, Musco, Musco, Velingker, and Zandieh (AKMMVZ19), which discretizes this problem via the leverage scores of the regression matrix.

Second, the authors utilize connections between spherical harmonics and zonal harmonics to enable the implementation of leverage score sampling of the regression matrix in question. This leverage score sampling result is novel.



**Strengths:**

### Result's Strength ###

I think the optimality of the stated result of the paper makes the paper mathematically strong.

----------------------------------------

# During the Rebuttal Phase #

The author's explanations, repeated readings of the paper, and Reviewer BcP8's review and questions have helped me understand and appreciate the paper better. I'm therefore raising the score from 4 to 6 and confidence from 2 to 3.


**Weaknesses:**

### Difficulty in reading. ###

I found the paper quite difficult to read though I could tell it was written well. I think this is simply because of the deeply technical nature of the problem and my unfamiliarity with the topic (at least to the level of generality of this paper) within the context of machine learning.

I am not sure if I have any concrete feedback to this end but due to this particular reason I feel this paper might be a much better fit (in terms of reaching a wide audience who'd actually understand and appreciate the results) at a mathematical/optimization journal. That said, I acknowledge, based on my unfamiliarity with the topic, that my judgement could be completely misplaced.

I'll be grateful to the authors if they could situate their work more in the context of machine learning. I'll also be happy to keep reading the submission and improving my understanding through the rebuttal period.

----------------------



**Questions:**

### Questions ###

1. Would the authors be able to situate their results in the context of machine learning? For instance, in lines 20-21, the potential applications mentioned seem to be leaning towards physics/astronomy; it would be quite interesting to see such applications in machine learning (e.g., published in prior ICML/NeurIPS conferences) and also some that might be somewhat more recent.

2. I'm actually slightly confused about the main result (Theorem 1 in the full version of the submission; lines 53-56 in the main_full.pdf) for the following reason: In "Randomized algorithms for matrices and data" by Mahoney (2011), we see that the (essentially optimal) sample complexity of least-squares regression is $d/\epsilon^2$, obtained by leverage score sampling. Here, $d$ can be thought of as the "intrinsic dimension" of the projection matrix in question; I'm surprised the result in the paper (Theorem 1 in the full submission) seems to have a better dependence of $\epsilon^{-1}$ on the sample complexity despite (seemingly) being a more general result. Would the authors be able to comment on this?

--------------------------

### Suggestions ###

1.  I think it would really help with readability if the authors could present "specific cases" of their statements wherever possible. As an example, the Definition 3, Equation 8 is essentially the definition of the sensitivities function (introduced by Langberg and Schulman; for a recent paper with this fact explicitly stated, see, for example, "Sharper Bounds for $\ell_p$ Sensitivity Sampling" by Woodruff and Yasuda); similarly, the minimum characterization of the leverage score function is essentially the one seen in Lemma 2 of "Uniform Sampling for Matrix Approximation" by Cohen, Lee, Musco, Musco, Peng, and Sidford). It would make it easier for readers to see these (possibly more familiar) statements first followed by the generalized versions stated in the paper.

---

> ### Author Rebuttal · Authors · 2023-08-09
>
> We greatly appreciate the reviewer's detailed and insightful comments.
>
> > `Would the authors be able to situate their results in the context of machine learning? For instance, in lines 20-21, the potential applications mentioned seem to be leaning towards physics/astronomy; it would be quite interesting to see such applications in machine learning (e.g., published in prior ICML/NeurIPS conferences) and also some that might be somewhat more recent.`
>
> Our main focus is recovering (unknown) functions defined in the sphere, a critical task in scenarios where rotational invariance is a fundamental property. In real-life machine learning applications, this property becomes very important as a foundational requirement for modeling 3D point clouds. Notable examples that appeared in top-tier machine learning conferences include molecular/atom systems, where understanding the underlying functions within a spherical context can significantly enhance predictive modeling and simulation accuracy [1,2,3,4]. Other examples are in the field of computer vision, specifically in the recognition, classification, and reconstruction of 3D objects [5,6,7,8].
>
> [1] Eickenberg, Michael, et al. "Solid harmonic wavelet scattering: Predicting quantum molecular energy from invariant descriptors of 3D electronic densities." NeurIPS 2017
>
> [2] Frank, Thorben, et al. "So3krates: Equivariant attention for interactions on arbitrary length-scales in molecular systems." NeurIPS 2022
>
> [3] Zitnick, Larry, et al. "Spherical channels for modeling atomic interactions." NeurIPS 2022
>
> [4] Liu, Yi, et al. "Spherical message passing for 3d molecular graphs." ICLR 2022
>
> [5] Gardner, James, et al. "Rotation-Equivariant Conditional Spherical Neural Fields for Learning a Natural Illumination Prior." NeurIPS 2022
>
> [6] Melnyk, Pavlo, et al. "Steerable 3D spherical neurons." ICML 2022
>
> [7] Gerken, Jan, et al. "Equivariance versus augmentation for spherical images." ICML 2022
>
> [8] Shakerinava, Mehran, and Siamak Ravanbakhsh. "Equivariant networks for pixelized spheres." ICML 2021
>
>
> > `I'm actually slightly confused about the main result (Theorem 1 in the full version of the submission; lines 53-56 in the main_full.pdf) for the following reason: In "Randomized algorithms for matrices and data" by Mahoney (2011), we see that the (essentially optimal) sample complexity of least-squares regression is d/eps^2, obtained by leverage score sampling. Here, d can be thought of as the "intrinsic dimension" of the projection matrix in question; I'm surprised the result in the paper (Theorem 1 in the full submission) seems to have a better dependence of eps^{-1} on the sample complexity despite (seemingly) being a more general result. Would the authors be able to comment on this?`
>
> In our proof of Theorem 1, we invoked Theorem 6.3 from the paper [Chen, Price'19]. Your intuition is correct that in order to achieve a subspace embedding guarantee for the design matrix in the regression problem with an approximation factor $\epsilon$ (i.e. to preserve all singular values of the gram matrix to within a factor of $1 \pm \epsilon$), it is generally necessary to have about $d \log d / \epsilon^2$ leverage score samples. The quadratic dependence on $1/\epsilon$ arises from the birthday's paradox, and the $\log d$ factor comes from the coupon-collector problem. The approximate regression results in Mahoney (2011) are derived from subspace embedding guarantee with error factor $\epsilon$.
>
> However, when approximately solving the regression problem, it is possible use fewer samples since we do not need to guarantee subspace embedding with an error parameter $\epsilon$. Instead, it suffices to have a subspace embedding that preserves the eigenvalues of the design matrix up to some constant factor, like $1 \pm 1/2$. For more precise details, please refer to the definition of "well-balanced sample" in Definition 2.1 of [Chen, Price'19] which is a sufficient condition for approximately solving linear regression.
>
> [Chen, Price'19]: "Active regression via linear-sample sparsification." COLT 2019.
>
>
> > `I think it would really help with readability if the authors could present "specific cases" of their statements wherever possible. ... It would make it easier for readers to see these (possibly more familiar) statements first followed by the generalized versions stated in the paper.`
>
> Your suggestion regarding enhancing readability is highly appreciated. To make it easier for readers to understand the concepts, we'll include references that they're familiar with. We'll follow your advice by adding specific cases to all our definitions. We agree with you that this will help readers start with recognizable statements before moving on to the generalized versions in our paper.

---

> > ### Comment · Reviewer_L1Mq · 2023-08-11
> > **Thank you so much! Requesting some clarification.**
> >
> > Dear authors,
> >
> > Thank you so much for that very detailed clarification. Your responses, plus Reviewer BcP8's very detailed review have helped me situate the problem a lot better.
> >
> > Working my way through the paper (and pattern-matching the objects to RandNLA concepts I know), I can appreciate the work more now.
> >
> > I have a small suggestion: It looks (from the appendix) that Lemma 1, 2, and 3 and Theorem 3 are all previously known classical results. I think it would be better to cite where they appeared in the statements of these lemmas/theorems in the main body itself, just to have a clear separation between known facts and ones you show.
> >
> > I was hoping for a clarification: What would you say is the key technical insight in your paper? Is it that the leverage scores of the projection operator are all constant, thereby enabling uniform sampling?
> >
> > I still wouldn't say I understand the paper entirely but I definitely see it better now. For this reason, I'm increasing my score; but I strongly feel that the paper needs a lot of additional writing (perhaps in the appendix) for it to be comprehensible to the general NeurIPS audience (and also for its results to be appreciated and adapted widely).
> >
> > Thank you again!

---

> > > ### Author Response · Authors · 2023-08-14
> > > **Response to Reviewer**
> > >
> > > Thank you very much for taking the time to consider our rebuttal response and for providing valuable feedback. We're pleased to hear that the additional clarification and Reviewer BcP8's insights have helped you better situate our result.
> > >
> > > Regarding your suggestion about citing the sources of Lemma 1, 2, 3, and Theorem 3, we appreciate your point and agree that it would enhance the clarity of our results and contributions. We will certainly make this adjustment in the revised version of the paper to provide a more cohesive presentation.
> > >
> > > In response to your question about the key technical insight of our paper, you've accurately captured one of our central contributions. The uniformity of leverage scores of the projection operator, which makes uniform sampling nearly optimal, is indeed a central element of our work. We will make this insight more explicit in the revised paper.
> > >
> > > We greatly appreciate your suggestion to include more explanatory content, especially in the appendix, in order to make our work more accessible to a broader audience. In our revised paper, we will take into account both your feedback and that of the other reviewers. We will also focus on providing more context and explanatory material in the appendix to ensure the broader audience can engage effectively with our findings.
> > >
> > > Once again, thank you for your thoughtful review and constructive feedback. We greatly appreciate your efforts in helping us improve the quality and accessibility of our paper. Your input is invaluable to us.

---

### Official Review · Reviewer_jKVP · 2023-07-03

**Soundness:** 4 excellent
**Presentation:** 2 fair
**Contribution:** 2 fair
**Rating:** 4
**Confidence:** 5

**Summary:**

A technique is proposed to recover spherical harmonic expansions for functions defined on a d-dimensional sphere from a set of function evaluations.  Spherical harmonic expansions are recovered by solving an optimizaiton problem via a kernal approach, which is accompanied by theoretical guarantees.  Numerical experiments demonstrate phase transitions close to the theoretical bounds.

**Strengths:**

A solid theoretical analysis is presented to derive a new approach to recovering spherical harmonic expansions from the evaluation of functions on the d-dimensional sphere.  It is shown that functions should be evalued uniformly randomly over the sphere.  The approach presented is accompanied by numerous theoretical results and guarantees.  Experimental results validate the theory presented, with phase transitions in success probabilities close to the theoretical bounds.

**Weaknesses:**

While the method is interesting, it does not seem that NeurIPS is the appropriate venue for this work.  While the field of deep learning on the sphere is an active area of research (e.g. [Cohen et al.](https://arxiv.org/abs/1801.10130)), this contribution would appear to be somewhat orthogonal to that body of literature.  No connection is made in the submitted manuscript beyond the final, somewhat cryptic, comment: "We believe our finding would appeal to the readership of the community".

Furthermore, the connection to previous literature of the topic of fast spherical harmonic transforms and efficient sampling on the sphere is very poor.  No reference is made to the central works of [Driscoll & Healy](https://www.sciencedirect.com/science/article/pii/S0196885884710086) and [McEwen & Wiaux](https://ieeexplore.ieee.org/document/6006544) and the follow-up articles of their groups.

The use of the term "near optimal" in the title and throughout the article is somewhat overstated.  The proposed method is "near" optimal up to a logarithmic factor.  I would typically expect near optimal to be up to a constant factor.

**Questions:**

How is the proposed method relevant for the field of deep learning on the sphere?  Can it offer a differentiable transform that can be integrated into approaches such as  [Cohen et al.](https://arxiv.org/abs/1801.10130)?

What is meant by the comment: "it is generally intractable to compute an orthogonal basis for the space of spherical harmonics, which renders the generalized Foureri series expansion in Lemma 2 primarily existential"?  This comment is repeated twice in the manuscript.
 Could this please be elaborated?

**Limitations:**

No special negative societal impacts.

---

> ### Author Rebuttal · Authors · 2023-08-09
>
> Thank you for your valuable feedback on our paper. We greatly appreciate your insights and suggestions.
>
> > `It does not seem that NeurIPS is the appropriate venue for this work. While the field of deep learning on the sphere is an active area of research (e.g. Cohen et al.), this contribution would appear to be somewhat orthogonal to that body of literature.`
>
> We appreciate your feedback on our work's relevance to deep learning on the sphere, inspired by research like Cohen et al. Although our work touches on some deep learning aspects, it's not exclusively focused on them. NeurIPS covers a broad range of research areas, with deep learning being just one facet among twelve distinct research domains highlighted in the NeurIPS 2023 Call for Papers. Our work aligns more closely with "Probabilistic methods," a specific area specified in the Call for Papers. We believe our research introduces a fresh perspective and valuable insights to the NeurIPS community. While we recognize potential concerns about its interdisciplinary nature, we firmly believe that our contributions can foster novel ideas and collaborations across research threads, ultimately enhancing the scientific landscape.
>
>
> > `No reference is made to the central works of Driscoll & Healy and McEwen & Wiaux and the follow-up articles of their groups.`
>
> Rest assured, we fully recognize the significance of these foundational works and in our final submission version, we will provide a comprehensive review of the key works by Driscoll & Healy and McEwen & Wiaux. Additionally, we will include references to the relevant follow-up articles to ensure we properly acknowledge their contributions.
>
>
> > `The use of the term "near optimal" in the title and throughout the article is somewhat overstated.`
>
> We want to highlight the common practice of describing "near optimal" performance up to a logarithmic factor, which is widely recognized in algorithmic literature.
> To provide some context, many prominent works in the field have adopted this perspective. For instance, the paper by Lin, Tianyi, Chi Jin, and Michael I. Jordan titled "Near-optimal algorithms for minimax optimization" (COLT 2020) employs the $\tilde{O}$ notation to discuss near-optimal solutions. Similarly, the study by Candès and Terence Tao titled "The power of convex relaxation: Near-optimal matrix completion" (IEEE Transactions on Information Theory 2010) employs this notation to describe the performance of their proposed approach.
> We believe that the use of "near optimal" to indicate optimality up to a logarithmic factor provides a more nuanced and accurate representation of our method's performance and aligns with the broader trend in algorithmic analysis.

---

> > ### Comment · Reviewer_jKVP · 2023-08-16
> > **Response to authors**
> >
> > Many thanks for the response to my queries.  I have reviewed the comments by all reviewers and all authors' responses.
> >
> > Thank you for the justification of the term "near optimal", which given its common use seems in line with the related literature.
> >
> > The article is technically very solid and clear (although in my second review I noticed the dagger operator in Algorithm 1 is not defined), however my initial assess of the manusciprt remains mostly unchanged.  I remain unconvinced that NeurIPS is the appropriate venue for this work.  While NeurIPS does indeed include a focus on probabilistic methods, I still do not view the article as a good fit under that topic.
> >
> > After reviewing the authors responses and other reviewer comments and responses I have raised by overall recommendation from 3 (Reject) to 4 (Borderline reject).
> >
> > I congratulate the authors on an excellent piece of work.  In my humble opinion, however, I do not believe NeurIPS is the approriate venue for this work.

---

### Official Review · Reviewer_6ewG · 2023-07-04

**Soundness:** 3 good
**Presentation:** 3 good
**Contribution:** 3 good
**Rating:** 6
**Confidence:** 3

**Summary:**

The paper studies the approximation of a function $f \in L_2(\mathbb{S}^{d-1})$ from its evaluations, via a degree-q spherical harmonic expansion. To this end, an efficient kernel regression based algorithm is proposed which recovers such a degree-q expansion of f, from the evaluations of f on $\mathbb{S}^{d-1})$. In particular, the number of evaluations needed scales nearly linearly in the dimension of the space of spherical harmonics of degree at most $q$. The main idea is to exploit connections between spherical harmonics and zonal harmonics, and the fact that the zonal harmonics are the reproducing kernels of the space of degree $l$ spherical harmonics. Some numerical simulations are provided on synthetic examples to demonstrate the performance of the algorithm.

**Strengths:**

1. The paper is written well overall with a well-defined problem statement, and a clear description of related work.


2. The results hold for any dimension $d$ which does not seem to have been handled previously. As noted in the related work, previous results typically applied to small, fixed values of $d$. In that respect, I think the results are quite strong.


**Weaknesses:**

1.  It’s a bit unclear to me whether the results hold only in the noiseless case, or is the method actually robust to noise. For instance, if we the function evaluations are corrupted with iid centred Gaussian noise, what can be said about the recovery error? The abstract mentions that the algorithm provides robust recovery, but I am not sure if this is what is proven.

2.  While I understand the main contributions are theoretical, are there any real examples on which the method can be evaluated? At the moment, experiments are only conducted on synthetical examples.


**Questions:**

1. In the optimization problem after line 42, isn’t the solution non-unique? If $g^*$ is a solution, then $g^* + g’$ for any g’ in the null space of the operator is also a solution? Also, isn’t f a solution of this optimization problem?

2. Just to clarify my understanding, my first thought was to simply do linear regression in the basis of the lower degree spherical harmonics. But I suppose this is intractable since computing such a basis is computationally hard for even moderate values of q as noted in the paper. Is this correct? And so, this is why we can only hope for an approximate solution of Problem 1?

3. In Definition 3, couldn’t we define the leverage function for any operator mapping $ L_2(\mathbb{S}^{d-1})$ to itself? Or is it specifically defined for $\mathcal{K}_d^{(q)}$?


**Limitations:**

I do not see the limitations discussed anywhere but this probably does not apply to this paper since it is essentially theoretical, and the sample complexity bounds are shown to be nearly optimal. Perhaps the running time which is super-quadratic in the number of samples could be improved upon?

---

> ### Author Rebuttal · Authors · 2023-08-09
>
> Thank you for your valuable feedback on our paper. Below we answer your concerns and questions.
>
> > `It’s a bit unclear to me whether the results hold only in the noiseless case, or is the method actually robust to noise. For instance, if we the function evaluations are corrupted with iid centred Gaussian noise, what can be said about the recovery error? The abstract mentions that the algorithm provides robust recovery, but I am not sure if this is what is proven.`
>
> In our paper, we consider a specific noise model to address the robustness of our method. We assume that the unknown function $f$, which we can query its values, is not necessarily a low-degree spherical harmonic and may contain high-degree components. These higher-degree components in the spherical harmonic expansion are treated as noise in our model.
>
> To clarify, the noise we consider in our study is different from the typical iid noise that can corrupt measurements of the function. Instead, we assume that the noise values are drawn from an underlying and unknown $L_2(S^{d-1})$ function. Under this noise model, our algorithm can successfully recover the function up to a $(1+\epsilon)$ factor of the noise's $2$-norm.
> Our results indeed hold under the above-specified noise model, which treats the higher degree components of $f$ as noise. Within this noise framework, we have proven the robustness of our algorithm.
>
>
> > `While I understand the main contributions are theoretical, are there any real examples on which the method can be evaluated?`
>
> Our method can be applied to any learning problem on the sphere. Spherical functions play a crucial role in problems with rotational-invariant property, where some real-world applications in machine learning is to recover 3D objective detection [1, 4, 5, 6], lighting estimation from images [2], and predicting atomic energies and forces [3].
>
> [1] Melnyk, Pavlo, et al. "Steerable 3D spherical neurons." ICML 2022
>
> [2] Gardner, James, et al. "Rotation-Equivariant Conditional Spherical Neural Fields for Learning a Natural Illumination Prior." NeurIPS 2022
>
> [3] Zitnick, Larry, et al. "Spherical channels for modeling atomic interactions." NeurIPS 2022
>
> [4] Gardner, James, et al. "Rotation-Equivariant Conditional Spherical Neural Fields for Learning a Natural Illumination Prior." NeurIPS 2022
>
> [5] Gerken, Jan, et al. "Equivariance versus augmentation for spherical images." ICML 2022
>
> [6] Shakerinava, Mehran, and Siamak Ravanbakhsh. "Equivariant networks for pixelized spheres." ICML 2021
>
>
> > `Just to clarify my understanding, my first thought was to simply do linear regression in the basis of the lower degree spherical harmonics. But I suppose this is intractable since computing such a basis is computationally hard for even moderate values of q as noted in the paper. Is this correct? And so, this is why we can only hope for an approximate solution of Problem 1?`
>
> You’re right about solving a linear regression in the basis of the low-degree spherical harmonics. While this approach could solve the problem, it still requires discretizing the linear regression to deal with basis vectors which are continuous functions.
> One of our main objectives is to minimize the number of samples needed to accurately recover the unknown function. However, solving the exact linear regression demands inner products between the basis vectors and the input function, leading to an infinite number of required samples from the input function.
>
> Furthermore, even though it's possible to construct a basis for spherical harmonics, as shown in Theorem 5.1 (https://arxiv.org/pdf/1304.2585.pdf), these basis functions are exceptionally complex. As a result, numerical calculations, even in moderate dimensions and with a moderate degree q, become quite challenging and intractable.
> On the other hand, our approach is based on kernel regression, which ensures numerical stability even in high dimensions and for large degree q. This makes our method more practical and easier to implement compared to using spherical harmonics.
>
> > `Can the leverage function be defined for any operator?`
>
> Yes, you are right the leverage function can be defined for any compact operator (please see Definition3 in https://arxiv.org/pdf/1812.08723.pdf ). We will modify our Definition 3 to make it clear that the leverage function is defined for any compact operator.

---

> > ### Comment · Reviewer_6ewG · 2023-08-16
> > **Replying to authors**
> >
> > Thank you for your response to my queries. I still have the following questions, it would be great if the authors could clarify them.
> >
> > 1. It would be helpful if the authors could answer Q1 in my review regarding the non-uniqueness of the solution of the optimization problem.
> >
> > 2. I am hesitant to consider the noise model in the paper as truly ``noise'' since the latter term is typically reserved for external stochastic/ adversarial noise in the samples. In that respect, I am not sure if it is correct to claim that the method is really robust to external noise. Moreover, it would be interesting to atleast discuss the technical difficulties encountered from a theoretical perspective for establishing this result. The experiments section could also demonstrate the effect of iid Gaussian noise on the performance of the method.
> >
> > 3. Thanks for your response to my question Q2. While I understand the drawback of linear regression from the computational perspective, its not clear to me why this would need infinite samples to work. I am drawing an analogy with what one does in the usual nonparametric regression setting for learning functions in $L_2[0,1]^d$ -- we can simply take a trigonometric basis (the first $m$ terms) and do finite dimensional linear regression.
> >
> > 4. Continuing point 3 above, I was wondering if the authors could comment on the precise running time of computing the first few (lower degree) basis functions. This would be relevant in terms of understanding the precise time complexity of implementing linear regression, and how it compares with the proposed method.

---

> > > ### Author Response · Authors · 2023-08-17
> > > **Response to Reviewer's questions**
> > >
> > > We truly appreciate your time and effort in evaluating our work. We have carefully considered your concerns and questions and would like to offer our explanations below:
> > >
> > > `1. Question regarding the non-uniqueness of the solution of the optimization problem`
> > >
> > > Yes, you are right. The optimization problem does not have a unique solution and the uniqueness of the solution is not required. In fact, as you mentioned, if $g$ is a solution then $g+g’$ for any $g’$ in the null space of $\mathcal{K_d}^{(q)} $ is also an optimal solution. Our aim is to find any (approximately) optimal solution $g$ and then we will project it onto the space of spherical harmonics of degree at most $q$ by considering $ \mathcal{K_d}^{(q)} g$, which will be (approximately) the spherical harmonics expansion of $f$. Also, $f$ itself is an optimal solution to the optimization problem in line 42.
> > >
> > > `2. I am hesitant to consider the noise model in the paper as truly ``noise'' since the latter term is typically reserved for external stochastic/ adversarial noise in the samples`
> > >
> > > Our noise model encompasses any perturbation introduced into the “input signal” prior to the recovery process, which includes adversarial perturbations as well. We believe that our noise model aligns with common assumptions made in adversarial noise models. In these models, it is typically assumed that an adversary introduces noise to the input signal "prior to" the recovery process, and in our setting the recovery process does include collection of samples from the input signal. By randomizing sampling positions, the recovery algorithm prevents capturing excessive noise energy. Yet, if an adversary could see the random pattern used by the recovery algorithm, then it can concentrate noise energy in sampled points to disrupt recovery.
> > >
> > > That being said, we've come to realize that we are able to upper-bound the norm of perturbations caused by iid Gaussian noise added to our measurements. Suppose that in Theorem 5 there are no higher degree spherical harmonics present in the expansion of the input function $f$, resulting in $f = f^{(q)}$. If we denote the noise vector as ${\bf e} \in R^s$ and the kernel matrix in Algorithm 1 by $ {\bf K} \in R^{s \times s} $, we can demonstrate that the perturbation's norm in the output $y$ of our algorithm (as defined in Theorem 5) caused by this noise is as follows:
> > >
> > > $ || y - f^{(q)} ||_{S^{d-1}}^2 = (1/s) \cdot {\bf e}^T {\bf K}^+ {\bf K} {\bf K}^+ {\bf e} $
> > >
> > > Now, note that $ {\bf e}^T {\bf K}^+ {\bf K} {\bf K}^+ {\bf e}$ is a nonnegative random variable with expected value $E[ {\bf e}^T {\bf K}^+ {\bf K} {\bf K}^+ {\bf e}] = {\tt tr} ({\bf K}^+ {\bf K} {\bf K}^+)$, thus by Markov’s inequality with 0.99 probability this random variable will be bounded by $O( {\tt tr} (K^+ K K^+) )$. Thus, the total perturbation to the output y is bounded by $ || y - f^{(q)} ||_{S^{d-1}}^2 \le O(1/s) \cdot {\tt tr} ({\bf K}^+ {\bf K} {\bf K}^+) $.
> > >
> > > Additionally, by considering the svd of ${\bf K}$ and ${\bf K}^+$ one can see that ${\tt tr} ({\bf K}^+ {\bf K} {\bf K}^+) = 1/\lambda_1 + 1/\lambda_2 + \ldots 1/\lambda_r$, where $\lambda_i$’s are the singular values of the kernel matrix ${\bf K}$. Now if we let ${\bf P}$ be the quasi-matrix defined in Theorem 4 then we have that the singular values of the kernel matrix $ {\bf K} = {\bf P}^* {\bf P} $ are equal to that of ${\bf P} {\bf P}^*$. On the other hand, using matrix Chernoff inequalities we can show that all singular values of the operator ${\bf P} {\bf P}^*$ approximate the singular values of the projection operator $ \mathcal{K_d}^{(q)} $ up to a constant factor. So we have $ {\tt tr} ({\bf K}^+ {\bf K} {\bf K}^+) = O( {\tt rank}( \mathcal{K_d}^{(q)} ) ) = O(\beta_{q, d})$. Finally, because $s \ge \Omega( \beta / \epsilon )$ this implies that:
> > >
> > > $|| y-f^{(q)} ||_{S^{d-1}}^2 \le \epsilon $.
> > >
> > > We will add a formal and precise version of the above proof sketch to the final version of our paper.
> > >
> > > We will answer questions 3 and 4 in a separate comment that will follow.

---

> > > ### Author Response · Authors · 2023-08-17
> > > **Continuation of response to reviewer's questions**
> > >
> > > `3. While I understand the drawback of linear regression from the computational perspective, it's not clear to me why this would need infinite samples to work. I am drawing an analogy with what one does in the usual nonparametric regression setting for learning functions in [0,1]^d we can simply take a trigonometric basis (the first m terms) and do finite dimensional linear regression.`
> > >
> > > We might be misunderstanding the reviewer’s question. Please inform us if the following explanation is addressing the right question. As we interpret it, the reviewer is proposing to construct a quasi-matrix $ Y$ which has $\beta_{q,d}$ columns made up of all basis functions for the space of spherical harmonics of degree $\le q$. Each column of Y is a spherical harmonic function and its columns together span the space of spherical harmonics. Now, our understanding is that you are suggesting to solve the subsequent linear regression problem:
> > >
> > > $\min_{x \in R^{\beta_{q,d}}}  || Y x - f ||_{S^{d-1}}^2$
> > >
> > > where $f$ is the input function. This is analogous to the least square problem we considered in line 42. Solving this least squares problem “exactly” using the normal equation requires computing $Y^* f$ which is nothing but the inner product of all our basis functions with the input function $f$. Essentially, we need to project the input function $f$ onto our basis functions.
> > >
> > > However, these basis functions are continuous, making the computation of these inner products reliant on knowing the value of $f$ across its entire domain. This essentially necessitates an infinite number of samples from the function $f$. It appears that addressing this regression problem would demand a discretization approach akin to the techniques we employed in our paper.
> > >
> > >
> > > `4. Continuing point 3 above, I was wondering if the authors could comment on the precise running time of computing the first few (lower degree) basis functions. This would be relevant in terms of understanding the precise time complexity of implementing linear regression, and how it compares with the proposed method.`
> > >
> > > Looking at Theorem 5.1 (https://arxiv.org/pdf/1304.2585.pdf), calculating the value of one basis function on a single point on the sphere requires calculating d trigonometric functions and raising them to powers up to $d^2$. One additionally needs to calculate values of $d$ Gegenbauer polynomials of various degrees on d different points. The overall time complexity of this is $O(q d^2)$. There are $ \beta_{q,d} $ basis functions in total so calculating all basis function values in a single point takes $O( q d^2 \cdot \beta_{q,d} )$ times. To solve the regression problem one needs to discretize using $s$ samples and so the total time to just compute discretized basis would be $O(q d^2 \cdot s)$.

---

> > > > ### Comment · Reviewer_6ewG · 2023-08-18
> > > > **Replying to authors response**
> > > >
> > > > Thank you for your reply. In point 3, I actually meant the following.
> > > >
> > > > Denote $u= [u_1,\dots,u_s]^{\top} \in \mathbb{R}^s$ to be the (noisy) samples of the unknown $f \in L_2(\mathbb{S}^{d-1})$ and let $\{y_1,\dots,y_{\beta_{q,d}}\}$ be an orthonormal basis of spherical harmonics. Denote $Y \in \mathbb{R}^{s \times \beta_{q,d}}$ to be the basis matrix where the $i^{th}$ column of $Y$ corresponds to the evaluations (at the $s$ sampling points) of the basis function $y_i$. Then we can solve the following least squares problem:
> > > >
> > > > $\min_{x \in \mathbb{R}^{s \times \beta_{q,d}}} ||u - Yx ||_2^2$
> > > >
> > > > and output $\hat{f} (\cdot) = \sum_{i=1}^{\beta_{q,d}} \hat{x}_i y_i(\cdot)$, where $\hat{x} = Y^{\dagger} u$. So if we can compute the matrix $Y$ efficiently, then there is no problem computationally. Note that the norm in the above problem is the $\ell_2$ norm.
> > > >
> > > > I am a bit confused about why this should not work since, from your reply to point 4 in my post, it seems that we can compute the matrix $Y$ efficiently. Could you please clarify this? Furthermore, I think what you consider in Algorithm 1 is to essentially replace $Y$ with a kernel matrix $K$, is this correct? If so, then is the benefit from a computational point of view?

---

> > > > > ### Author Response · Authors · 2023-08-19
> > > > > **Response to reviewer**
> > > > >
> > > > > Thanks for clarifying your question. Yes, what you proposed works and is indeed similar to our algorithm 1. In fact, your approach is equivalent to ours because, leveraging certain properties of linear regression, we can transform your method into our kernel regression as follows:
> > > > >
> > > > > First, consider your proposed least squares problem:
> > > > >
> > > > > $\min_{x \in \mathbb{R}^{\beta_{q,d}}} || Yx - u ||_{S^{d-1}}^2$
> > > > >
> > > > > There must exist an optimal solution for this problem which lies within the row span of Y. Therefore, the optimal solution, denoted as $x_{opt}$, can be expressed as $x_{opt} = Y^T z$ for some $z \in R^s$. Thus, your regression problem is equivalent to:
> > > > >
> > > > > $\min_{z \in \mathbb{R}^s} || YY^Tz - u ||_{S^{d-1}}^2$
> > > > >
> > > > > Now, let's examine the matrix $YY^T$, where the $(i, j)$-th entry is given by:
> > > > >
> > > > > $\[Y Y^T\](i, j) = \sum_{t = 1}^{\beta_{q, d}} y_t(\sigma_i) y_t(\sigma_j)$
> > > > >
> > > > >
> > > > > Here, $\sigma_1, \sigma_2, \ldots, \sigma_s$ represent random sampling points on the unit sphere. It follows from the Addition Theorem (refer to our Theorem 3 and eq(10)) that:
> > > > >
> > > > > $\[Y Y^T\](i, j) = \sum_{\ell=1}^q \frac{\alpha_{\ell,d}}{|S^{d-1}|}  P_d^{\ell} ( \langle \sigma_i, \sigma_j \rangle ) = s \cdot K_{i,j}$
> > > > >
> > > > > So, your regression problem can be equivalently expressed as:
> > > > >
> > > > > $\min_{z \in \mathbb{R}^s} || s \cdot Kz - u ||_{S^{d-1}}^2$
> > > > >
> > > > > We can factor out the constant '$s$' to obtain:
> > > > >
> > > > > $\min_{z \in R^s} s \cdot  || K z - u/\sqrt{s} ||_{S^{d-1}}^2$
> > > > >
> > > > > This expression is exactly equivalent to our Algorithm 1, particularly noting the scaling by $1/\sqrt{s}$ in line 5 of the algorithm, where we set ${\bf f} = u/\sqrt{s}$.
> > > > >
> > > > > However, if you were to solve the regression problem in its original form, the runtime would be slower. This is because calculating an entry of our kernel matrix involves calculating the value of a degree-$q$ polynomial, taking time on the order of $q + d$. In contrast, computing the basis functions $y_t(\cdot)$ at a random point takes time on the order of $qd^2$, as we discussed earlier.

---

### Official Review · Reviewer_7fua · 2023-07-06

**Soundness:** 4 excellent
**Presentation:** 3 good
**Contribution:** 3 good
**Rating:** 6
**Confidence:** 3

**Summary:**

To develop kernel regression based algorithm to recover degree-q expansion of f \in L^2(S^{}d-1| - by only evaluating on uniformly sampled points on S^{d-1}.


**Strengths:**

The ideas used in this paper is deeply technical and ideas are complicated. It first re-formulates the problem as least squares regression and then uses the sampling based on leverage scores to decide which samples to pick. Then the rest of the paper is proving the estimated bound is satisfied.

**Weaknesses:**

motivate the problem - how it could be applied in real life

** after rebuttal -- thank you for providing all these papers regarding contributions of this workstream in terms of application side. I am happy to increase my score regarding soundness of the paper.

**Questions:**

-

**Limitations:**

-

---

> ### Author Rebuttal · Authors · 2023-08-09
>
> > `Motivate the problem - how it could be applied in real life?`
>
> Many thanks for your valuable feedback on our paper.
> We understand the importance of motivating the problem and highlighting its real-life applications. In response to your comment, we would like to elaborate on the practical significance of the problem we address and its potential applications.
>
> Our main focus is recovering (unknown) functions defined in the sphere, a critical task in scenarios where rotational invariance is a fundamental property. In real-life applications, such as machine learning, this property becomes very important as a foundational requirement for modeling 3D point clouds. Notable examples that appeared in top-tier machine learning conferences include molecular/atom systems, where understanding the underlying functions within a spherical context can significantly enhance predictive modeling and simulation accuracy [1,2,3,4]. Other examples are in the field of computer vision, specifically in the recognition, classification, and reconstruction of 3D objects [5,6,7,8].
>
> [1] Eickenberg, Michael, et al. "Solid harmonic wavelet scattering: Predicting quantum molecular energy from invariant descriptors of 3D electronic densities." NeurIPS 2017
>
> [2] Frank, Thorben, et al. "So3krates: Equivariant attention for interactions on arbitrary length-scales in molecular systems." NeurIPS 2022
>
> [3] Zitnick, Larry, et al. "Spherical channels for modeling atomic interactions." NeurIPS 2022
>
> [4] Liu, Yi, et al. "Spherical message passing for 3d molecular graphs." ICLR 2022
>
> [5] Gardner, James, et al. "Rotation-Equivariant Conditional Spherical Neural Fields for Learning a Natural Illumination Prior." NeurIPS 2022
>
> [6] Melnyk, Pavlo, et al. "Steerable 3D spherical neurons." ICML 2022
>
> [7] Gerken, Jan, et al. "Equivariance versus augmentation for spherical images." ICML 2022
>
> [8] Shakerinava, Mehran, and Siamak Ravanbakhsh. "Equivariant networks for pixelized spheres." ICML 2021

---

### Official Review · Reviewer_BcP8 · 2023-07-27

**Soundness:** 4 excellent
**Presentation:** 4 excellent
**Contribution:** 3 good
**Rating:** 7
**Confidence:** 4

**Summary:**

Consider the $d$-dimensional unit sphere $\mathbb{S}^{d-1}$, and any function $f:\mathbb{S}^{d-1}\rightarrow\mathbb{R}$ defined on the sphere with bounded L2 norm $\|\|f\|\|_{\mathbb{S}^{d-1}}$. This function $f$ can be evaluated at any point $\vec w \in \mathbb{S}^{d-1}$ on the sphere, but it is expensive to evaluate $f$ so we wish to do this as little as possible.

The goal of this paper is to recover a near-optimal polynomial $\tilde f$ approximation to $f$, where $\tilde f$ is constrained to be a multivariate polynomial where the sum-of-degrees of each term is at most $q$. (For example, the sum-of-degrees of each term in $\tilde f(x,y,z) = x^2y^3 + z + x^5$ is at most 5).

Letting $\beta_{d,q}$ denote the number of free parameters in any $d$-variate polynomial with terms whose sum-of-degrees is at most $q$, this paper shows that $O(\beta_{d,q} \log \beta_{d,q} + \frac{\beta_{d,q}}{\varepsilon})$ evaluations of $f$ chosen uniformly at random on the sphere suffice to recover such a near-optimal $f$. This is proven in three parts:
1. The leverage function associated with this polynomial recovery problem is **exactly** the uniform distribution on the sphere
2. The randomized linear algebra toolkit shows this near-linear sample complexity suffices
3. A small kernel ridge regression problem can be solved to recover this $\tilde f$ from uniform random samples on the sphere

They also prove an $\Omega(\beta_{d,q})$ sample complexity lower bound. Lastly, they include some synthetic experiments.

**Strengths:**

The paper is a nice application of well understood tools from randomized linear algebra (RandNLA) to spherical harmonics, which has not been explored by the RandNLA literature afaik. The real core strength of this paper is its novelty in connecting these two literatures in a simple and elegant way.

Simplicity and elegance really are words that mark this paper. Almost everything is extremely clear and well written. There are basically no typos even! The flow of logic in the paper is very clear, the problem they solve seems important in the spherical harmonics literature (though I'm no expert in that domain), and the proofs are even pretty clean. To that last point, the theorems in this paper are proven either by very clean techniques that are now standard in the RandNLA literature, or by using well established classical facts about polynomials and spherical harmonics. I verified a decent amount of the math, and it all struck me as rather nice and clean.

Since I'm not an expert in spherical harmonics, I can't perfectly speak to the significance of the result. Taking the authors' words at face value, it seems that prior work in spherical harmonics did not use such simple algorithms (Kernel Ridge Regression) and did not achieve optimal sample complexities.

The paper may not carry a huge amount of new technical ideas to get their sample complexity and simple algorithm, but this result would only exist if someone who knew enough about both spherical harmonics and RandNLA decided to sit down and figure out if this all works together. For that view of novelty, in addition to the simplicity and elegance of the paper, I recommend accepting this paper.

The experiments are perfectly fine for a theoretical paper. Nothing particularly strong about them, but nothing I'm left looking for.

**Weaknesses:**

The novelty, simplicity, and elegance is strong. But, the proof techniques are not especially novel. The proof techniques are standard relationships between operators and algorithms as explored by the RandNLA community for a good few years now.

The only proof that doesn't seem directly tied to something already proven in the RandNLA literature is the proof that the leverage function for the regression problem is uniform on the sphere. Even then, this claim is pretty intuitive, since the problem statement is rotationally invariant. There's no reason for a sampling algorithm to care more about one point of the sphere than another. (This is in contrast to learning on an interval, where an algorithm may prefer to sample near the edges of the interval.) So, even this most novel proof is not terribly surprising, and the proof is short and simple.

That said, I hesitate to really call this a "weakness". This isn't clearly a downside of the paper. While it's certainly nice for a paper to overcome new technical issues and proof difficulties, it's also nice for a paper to show that spherical harmonics smoothly fit into the RandNLA framework.

All this is to say that the technical weight of the paper truly is in understanding two literatures and connecting them. The effort to connect the literatures seems to be low, but that's given that someone actually understands both literatures well. I'd say this overall comes out as a slight strength for the paper, but it's certainly a tradeoff in the reviewing process.

**Questions:**

I list minor questions, typos, and recommended edits here. These are all soft recommendations, adapt whatever you want to.

1. [Line 53] Consider mentioning a failure probability here.
1. [Line 87] Replace "this paper" with "[GMMM21]" since the language is a bit too ambiguous at a glance.
1. [Line 173] Is this not a bijection? At least for traditional least squares regression, this seems to be a bijection? RandNLA people tend to use the former form, so it's nice to see if the RandNLA form is exactly equivalent to the form of Problem 2.
1. [Line 213] Consider removing the $\cdot$ between P and v? This notation feels a bit clunky and odd to me?
1. [Line 220] Cite [AKM+] or [SA] or [CP from COLT19] here, or some other paper that draws a connection between the semi-infinite regression and kernel ridge regression. This section reads like the idea of the kernel trick here is a new contribution.
1. [Line 226] Add "Then," before "Algorithm 1"
1. [Line 236] Discuss why this isn't trivial that we need $\beta_{d,q}$ samples to learn $\beta_{d,q}$ parameters in a polynomial. Certainly, when we're looking at interpolating a polynomial on the real line, we need at least $q+1$ samples to learn a degree $q$ polynomial exactly. This feels similar to the argument made in this lower bound, but I don't understand what the rest of the formalization is needed for.
1. [Line 250] Replace "accurately" with "exactly" (I think that's more technically accurate?)
1. [Lines 263-274] I got really lost here. I think that If I'd read the lower bound of AKM, then I might be able to stitch together the parts of this lower bound, but the reason any of this construction is made is unclear to me. The notation is also pretty hard to track. I don't really get what the construction is trying to get at. I don't really see why the top rows of Q need to span the queries made so far. This all feels odd to me.
1. You don't need to do this. But, if you want to, I think you could make a lower bound of $\Omega(\frac1\varepsilon)$ by following the technique in Section 5.4 of [here](https://arxiv.org/pdf/2211.06790.pdf). It'd be interesting to see if this technique holds on the sphere, because at a glance it seems like it should. It'd give an overall lower bound of only $\Omega(\beta_{d,q} + \frac1{\varepsilon})$ though, which isn't the most compelling rate.

---

> ### Author Rebuttal · Authors · 2023-08-09
>
> Many thanks for your valuable feedback on our paper. Below we answer your questions.
>
>
> >`Discuss why this isn't trivial that we need \beta_{d,q} samples to learn \beta_{d,q} parameters in a polynomial.`
>
>
> You're absolutely correct in pointing out that there are $\beta_{d,q}$ degrees of freedom, thus any deterministic algorithm that reconstructs such polynomials needs at least $\beta_{d,q}$ samples. Our lower bound proof is showing that even a "randomized" algorithm that succeeds with only a constant probability, needs to take $\beta_{d,q}$ samples. Since our upper bound is established using a randomized algorithm, it was crucial to complement it with a randomized lower bound to create a well-rounded and balanced analysis of the problem.
>
> >`I think that If I'd read the lower bound of [AKM+], then I might be able to stitch together the parts of this lower bound, but the reason any of this construction is made is unclear to me. The notation is also pretty hard to track. I don't really get what the construction is trying to get at. I don't really see why the top rows of Q need to span the queries made so far. This all feels odd to me.`
>
>
> We apologize for any confusion caused by the clarity of our lower-bound section. The limited space may have contributed to the lack of detailed explanations.
> To provide further clarification, our hard instance is based on a random vector ${\bf v}$ following an isotropic Gaussian distribution in dimension $\beta_{d,q}$. In line 273, our aim is to demonstrate that if an algorithm reconstructs a function $\tilde{f}^{(q)}$ using only $r <  \beta_{d,q}$, then even after conditioning on the samples observed by the algorithm, vector ${\bf v}$ will still possess at least one degree of freedom and will not be entirely deterministic. This intuitively holds true because ${\bf v}$ consists of $ \beta_{d,q}$ independent random Gaussian entries. Thus, when conditioning on $r < \beta_{d,q}$ samples taken by an algorithm, the conditional value of ${\bf v}$ will remain random, allowing us to extract at least one Gaussian random variable from it using an orthonormal transformation denoted as $Q^r$.
> We will provide additional details and improve the notation for better readability.
>
>
> > `Regarding minor typos:`
>
> Thanks for pointing them out. We will address them all and implement your recommended edits.

---

> > ### Comment · Reviewer_BcP8 · 2023-08-10
> > **Thanks for you response**
> >
> > Hi, thanks for the response!
> >
> > If it's trivial that at least $\beta_{d,q}$ samples are deterministically needed, why doesn't Yao's Minimax Principle imply that any randomized algorithm also needs at least $\beta_{d,q}$ samples? (Admittedly, I'm sometimes a bit unclear about when Yao's can apply, but this seems like such a setting, and if it applies then Yao's would require a very short proof).

---

> > > ### Author Response · Authors · 2023-08-11
> > > **Response to reviewer's question**
> > >
> > > Many thanks for your prompt response.
> > >
> > > By Yao’s minimax principle, we can assume that the recovery algorithm is deterministic and requires constant probability recovery over a random ensemble of input functions, instead of considering a randomized algorithm and requiring constant probability recovery for “any” fixed given function in the ensemble.
> > >
> > > In light of this, our objective is to demonstrate that any deterministic algorithm aiming to recover a constant fraction of functions in an input function ensemble must need a minimum of $\beta_{d,q}$ samples.
> > >
> > > Now let us write precisely the degrees of freedom argument:
> > >
> > > `degrees of freedom argument:` In order to specify a spherical harmonic of degree <= q unambiguously, one needs to know the values of $\beta_{d,q}$ free parameters. This implies that any deterministic algorithm seeking to recover “all” inputs with probability 1 must utilize at least $\beta_{d,q}$ samples.
> > >
> > > Now recall that our aim is to prove that a deterministic recovery algorithm with a constant success probability over a random ensemble of input functions requires a minimum of $\beta_{d,q}$ samples. It's important to recognize that the degrees of freedom argument does not preclude the potential existence of an algorithm capable of recovering only a “constant fraction” of inputs within the entire ensemble using fewer than $\beta_{d,q}$ samples. It's worth noting that the requirement for a deterministic algorithm to achieve recovery for a constant fraction of input functions from the ensemble is a strictly weaker criterion compared to the more stringent demand of recovering “all” possible inputs.
> > >
> > > Therefore, one needs to analyze a random hard input distribution in order to prove the lower bound holds for even an algorithm with a constant success probability. Hopefully, our explanation clarifies our lower-bound results. Please let us know if we can provide further clarification.

---

### Decision · Program_Chairs · 2023-09-21

**Decision:**

Accept (poster)

**Comment:**

This paper uses randomized linear algebra (specifically leverage scores sampling) to solve near-optimally the natural problem of fitting a function on the sphere with spherical harmonics using a limited number of samples. Most reviewers felt the work studies and interesting problems, contains a strong theoretical result (not the most novel, but novel enough) and is reasonably well presented, We hope the authors take into account suggestions for how to improve clarity and accessibility to the broader NeurIPS audience when preparing their final version of the paper.